# Histone H3K4me3 modification is a transgenerational epigenetic signal for lipid metabolism in *Caenorhabditis elegans*

Qin-Li Wan [1,2,3,6], Xiao Meng[1,2,6], Chongyang Wang [1,2], Wenyu Dai [1,2], Zhenhuan Luo [1,2], Zhinan Yin [1,2], Zhenyu Ju [4], Xiaodie Fu[1,2], Jing Yang [1,2], Qunshan Ye [1,2], Zhan-Hui Zhang[5] & Qinghua Zhou [1,2 ✉]

As a major risk factor to human health, obesity presents a massive burden to people and society. Interestingly, the obese status of parents can cause progeny's lipid accumulation through epigenetic inheritance in multiple species. To date, many questions remain as to how lipid accumulation leads to signals that are transmitted across generations. In this study, we establish a nematode model of *C. elegans* raised on a high-fat diet (HFD) that leads to measurable lipid accumulation, which can transmit the lipid accumulation signal to their multigenerational progeny. Using this model, we find that transcription factors DAF-16/FOXO and SBP-1/SREBP, nuclear receptors NHR-49 and NHR-80, and delta-9 desaturases (*fat-5*, *fat-6*, and *fat-7*) are required for transgenerational lipid accumulation. Additionally, histone H3K4 trimethylation (H3K4me3) marks lipid metabolism genes and increases their transcription response to multigenerational obesogenic effects. In summary, this study establishes an interaction between a network of lipid metabolic genes and chromatin modifications, which work together to achieve transgenerational epigenetic inheritance of obesogenic effects.

[1] The Sixth Affiliated Hospital of Jinan University, Jinan University, Dongguan 523560 Guangdong, China. [2] The Biomedical Translational Research Institute, Faculty of Medical Science, Jinan University, Guangzhou 510632 Guangdong, China. [3] Department of Pathogen Biology, School of Medicine, Jinan University, Guangzhou 510632, China. [4] Key Laboratory of Regenerative Medicine of Ministry of Education, Institute of Aging and Regeneration Medicine, Jinan University, Guangzhou, Guangdong 510632, China. [5] The First Affiliated Hospital, Jinan University, Guangzhou, Guangdong 510632, China. [6]These authors contributed equally: Qin-Li Wan, Xiao Meng. ✉email: gene@email.jnu.edu.cn

Metabolic diseases are the result of a combination of genetic and environmental effects. Common environmental factors affecting metabolic diseases include nutrition, microbiome, exercise, and individual behaviors[1–4]. Interestingly, studies in animal models and humans have indicated that parents' experience can not only influence their own health but also affect their descendants, initiating a pass-down of disease risk across generations. For example, the parental experiences before and during pregnancy, are now recognized as an important factor in the incidences of obesity and type II diabetes (T2D)[5]. In modern societies, many people live in a condition of overnutrition that causes a high prevalence of obesity, and thus a variety of metabolic diseases such as T2D, hypertension, and coronary heart disease. Therefore, it is important to study whether and how obesity induces multigenerational inheritance, and to dissect the underlying molecular mechanisms.

The nematode worm *Caenorhabditis elegans* is a relevant and tractable model for the study of multigenerational obesogenic effects. Nematode genes for lipid metabolism are mostly conserved in mammalian models, as core lipid metabolic pathways in *C. elegans* have homologs in humans. The known transgenerational epigenetic inheritance (TEI) mechanism is also conserved across species. For example, histone modifications such as methylation of H3K4, H3K9, and H3K27, and inherited small RNAs are involved in TEI among multiple species in different scenarios[6–9]. Recently, some progress has been made in the understanding of TEI using nematodes as a model. For example, *C. elegans* learn to avoid pathogens and transmit the memory to their progeny for up to four generations[10–12]. An artificial environment such as Bisphenol A (BPA) exposure can cause transgenerational inheritance, with one-generational parental (P0) exposure of BPA leading to a decrease of H3K9me3 and H3K27me3 for five generations[8,13]. The memory of mitochondrial stress was reported to be transgenerationally inherited[14,15]. In our previous study, we found that hormetic heat stress-induced survival advantage, which had been reported in another study[16], could be passed down to progeny through histone H3K9me3 and DNA $N^6$-mA modifications[17]. A recent study reported that H3K9me2 was involved in TEI induced by heat shock[18].

The question of whether, and if so how, TEI plays a role in lipid metabolism has attracted the attention of researchers in recent years. For example, it was reported that the TEI obesogenic effects of sulfomethoxazole are associated with histone H3K4me3 modification[19]; another study indicated that a reduction in lipid accumulation induced by benzylisoquinoline can be transmitted to progeny[20]. However, the detailed mechanisms of the underlying multigenerational obesogenic effects remain largely unknown. For example, how are genes related to fat metabolism linked to chromatin modifiers to achieve the lipid metabolism-associated TEI? In this study, we use a high-fat diet (HFD, egg yolk) to feed *C. elegans* to establish a lipid accumulation model. Using this model, we characterize the critical factors functioning for "implementation" or "transmission" in TEI of lipid accumulation in *C. elegans*.

## Results

**Lipid accumulation from the HFD exposure can be transmitted across generations**. Supplementation of different lipids, including oleic acid (OA) or palmitoleic acid (PA), in food, can significantly increase fat accumulation in *C. elegans*[21,22]. Similarly, using Oil-Red-O (ORO) staining, we found that the ORO level was obviously elevated when animals were fed with OA, PA, cholesterol, or egg yolk compared with the OP50 medium, and that supplementation with egg yolk showed the most significant lipid accumulation (Fig. 1a). Therefore, in the subsequent assays, we used egg yolk as a HFD to feed *C. elegans* and established a fat accumulation model. To test whether the HFD-induced lipid accumulation phenomenon can be transmitted to the descendants raised on OP50, we randomly selected one wild-type hermaphrodite worm and obtained a population with the same genetic background through its self-fertilization. Then we fed wild-type worms with HFD and bleached the mothers to obtain F1, F2, and Fn generations (Fig. 1b). We found that the ORO levels were significantly increased, in both HFD-fed parents and their F1 and F2 progeny recovered on normal OP50 (Fig. 1c, d), suggesting that HFD can induce a multigenerational epigenetic inheritance phenotype on descendants from a single exposure of the P0 generation. Considering that the F1 generation may have been exposed to HFD while still inside the mother, in which case the obesogenic effect in the F1 worms would be considered intergenerational inheritance instead of transgenerational inheritance. Therefore, we fed P0 animals with HFD from L1 to L4 larvae, then transferred them to nematode growth medium (NGM) plates with normal OP50 to prevent F2 primordial germ cells from exposure. We found that pre-exposure of P0 worms to HFD until L4 larvae could also induce the lipid accumulation of their naïve F1 and F2 progeny (Fig. 1e), suggesting that HFD can induce a TEI obesity phenotype. Notably, animal populations in these assays were synchronized using L1 diapause. One concern is that the synchronization procedure might cause starvation in L1, which would influence the metabolism of animals and affect transgenerational gene expression[23]. Therefore, to ease our concern, we compared the TEI effects of worms with or without synchronization to find that synchronization did not affect TEI in our settings (Fig. 1h).

Different consecutive generations of exposure to the same environmental stress can enhance animals' adaptation to stress for different generations of descendants[15,24,25]. In our heritable obesity effect model, we found that F2 populations exhibited a measurable but minor lipid accumulation phenotype. In order to find if a longer exposure might enhance F2s' lipid accumulation, we exposed multiple consecutive generations of animals to HFD. We found that the exposure of four consecutive generations of animals (P0, F1, F2, and F3), three (F1, F2, and F3), two (F2, and F3), or one (F3), did not affect lipid accumulation of their recovered F4 or F5 descendants (Fig. 1f–g and Supplementary Fig. 1f–g). In addition, we observed that lipid accumulation induced by OA or PA can also transmit to their naïve progeny (Supplementary Fig. 1a–c). Altogether, these results demonstrated that exposure of HFD induces a transgenerational obesity effect.

We next wanted to determine how long a parent must be exposed to HFD to transmit lipid accumulation signal to progeny. For this, we fed P0 animals with HFD at the L1 stage for 12 or 24 h, and then transferred them to NGM plates with normal food. The results showed that the ORO levels were significantly increased in HFD-fed parents and their recovered F1 progeny, although to a lesser extent than worms from parents fed with HFD for longer periods, but not F2 progeny (Supplementary Fig. 1d, e), suggesting that parents need to be fed HFD for enough long time before they can show a TEI phenotype. Transgenerational information has been shown to be passed down through both male and female germline[17]. To determine the contributions of the sperm and oocytes in transmitting information of lipid accumulation, we conducted reciprocal mattings and analyzed lipid levels of cross progeny (Fig. 1i). Our results showed that both females fed with HFD and males fed with HFD transmitted information of lipid accumulation to the next generation, suggesting a role of both sperm and oocytes in the transgenerational obesity effect.

It is reported that lipid metabolism and chromatin modifications are associated with the regulation of lifespan[26–28]. In our

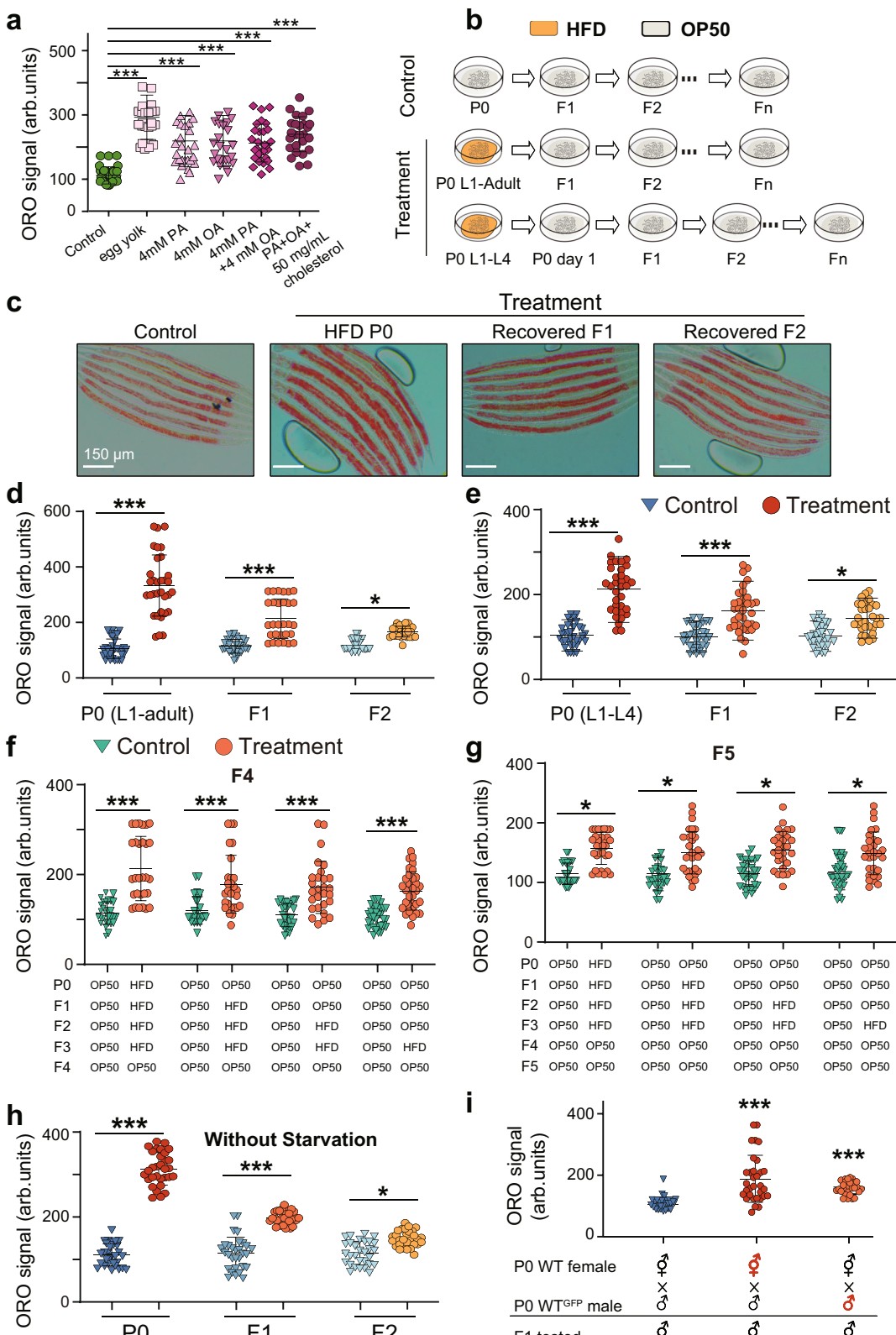

settings, we found that animals fed with egg yolk increased lipid accumulation. We wondered whether egg yolk affected the lifespan of *C. elegans*. Lifespan analyses revealed no difference between worms fed with HFD and OP50, suggesting that egg yolk did not affect the survival of *C. elegans* (Supplementary Fig. 2).

**HFD-induced transgenerational inheritance is mediated by nuclear receptors NHR-49, NHR-80, and transcription factors SBP-1 and DAF-16.** To dissect the molecular mechanisms underlying HFD-induced TEI, we evaluated the contribution of several nuclear receptors and transcription factors that play essential roles in lipid metabolism in *C. elegans*. SBP-1, a sterol regulatory element-

**Fig. 1 HFD-induced transgenerational lipid accumulation. a** Quantification of ORO staining of wild-type (N2) treated with different lipid foods ($n \geq 28$ worms per condition). PA palmitoleic acid, OA oleic acid. **b** Experimental scheme. **c, d** Transgenerational inheritance of lipid accumulation induced by HFD (egg yolk) in wild-type *C. elegans* ($n \geq 28$ per condition), scale bar = 100 μm. **e** Transgenerational inheritance of lipid accumulation induced by HFD in wild-type *C. elegans*. Worms were fed with HFD from L1 to L4 larvae before they were transferred to normal food until day 1 adulthood. Embryos collected by bleaching to obtain F1 animals that were fed with OP50. F2-Fn generations were obtained using the same protocol ($n \geq 34$ per condition). **f, g** Quantification of ORO staining of F4 or F5 wild-type (N2) from parents exposed to HFD four (P0, F1, F2 and F3), three (F1, F2 and F3), two (F2 and F3) and one (F3) generation of animals ($n \geq 28$ per condition). **h** Quantification of ORO staining of F1 or F2 from parents fed with HFD and without synchronization-caused starvation ($n \geq 30$ per condition). **i** Lipid accumulation information induced by HFD could be transmitted through both male and female germlines. Quantification of ORO staining of recovered F1 progeny of fathers fed with HFD or mothers fed with HFD ($n \geq 30$). For **a** and **d–i**, graph data are presented as mean ± SD (arb. units: arbitrary units), statistical analyses were performed by unpaired two-tailed Student's *t*-test and ANOVA analysis; ***$p < 0.001$ and *$p < 0.05$. Source data are provided as a Source Data file.

binding protein, is a crucial transcription factor governing fat metabolism[21]. NHR-49, a functional homolog of mammalian peroxisome proliferator-activated receptor alpha, and NHR-80, a homolog of mammalian hepatocyte nuclear factor 4, are both important nuclear hormone receptors involved in the control of fat consumption and fatty acid composition in *C. elegans*[29–32]. Forkhead transcriptional factor DAF-16/FOXO, a central downstream effector of the insulin/insulin-like growth factor signaling pathway, is a critical and well-conserved metabolic regulator[33,34]. Our results demonstrated that the elevated fat level induced by HFD in P0 was not affected in a *sbp-1* loss-of-function (lof) mutant (Fig. 2a), an *nhr-80* lof mutant (Fig. 2b) or an *nhr-49* lof mutant (Fig. 2c). However, lipid accumulation was abrogated in F1 or F2 descendants of these mutants. By contrast, the elevated lipid phenotype induced by HFD was abolished in P0 parents as well as in their recovery F1 and F2 progeny in the *daf-16* lof mutant (Fig. 2d). These results indicated that SBP-1, NHR-49 and NHR-80 work in the F1 generation to mediate TEI of fat accumulation induced by HFD; whereas DAF-16 not only acts in the F1 generation, but also functions for the execution of the lipid metabolism response in the P0 generation. To characterize the relationship of DAF-16, NHR-49, NHR-80 and SBP-1 in F1 generation, we performed epistasis analyses. We found that overexpressing DAF-16 in *nhr-49, nhr-80* and *sbp-1* mutant could not suppress the phenotype of *nhr-49, nhr-80* and *sbp-1* lof mutant (Supplementary Fig. 3a–c). Here, we applied *sbp-1* RNAi in the background of overexpressing DAF-16 to analyze lipid level as we failed to obtain *daf-16* (OE); *sbp-1* (lof) double mutant by numerous matings. These results demonstrated that *daf-16, sbp-1, nhr-49* and *nhr-80* function in parallel pathway during the stress of lipid accumulation in F1 generation.

As *daf-16* is responsible for HFD-induced lipid accumulation in P0 generation, to confirm the role of *daf-16* in the recovered F1, we used RNAi to knock down *daf-16* exclusively in F1 derived from P0 fed with HFD. We found that specific knockdown of *daf-16* in F1 also abrogated the elevated fat level (Fig. 2e, f). Moreover, we also performed similar analyses using *nhr-49, nhr-80* and *sbp-1* RNAi. We found that silencing *nhr-49, nhr-80* and *sbp-1* in F1 progeny also abolished their lipid accumulation (Supplementary Fig. 4a–e), suggesting that *daf-16, nhr-49, nhr-80* and *sbp-1* act in the progeny to increase lipid level.

As a key regulator, DAF-16 responds to environmental stress by activating a series of target genes[34]. We found that the mRNA level of the target genes of DAF-16 (*sod-3* and *dod-3*) was significantly upregulated in P0 fed with HFD or their naïve F1 progeny (Supplementary Fig. 5c). The expression of SOD-3::GFP was consistent with the mRNA level of *sod-3* (Supplementary Fig. 5a, b). Altogether, these results indicated that DAF-16 functioned in response to alter lipid metabolism induced by HFD in P0 and their recovered progeny.

To more definitively dissect whether the role of *daf-16, nhr-49, nhr-80* and *sbp-1* is to implement regulation of lipid level (an "executor"), to transmit the heritable memories (a "transmitter"), or both, we used RNAi to silence *daf-16, nhr-49, nhr-80* and *sbp-1* exclusively in the P0 generation, and then analyzed the fat level of F1 generation raised on OP50 (Fig. 2g). As reported in our recent study[17], if a gene is a transmitter, the silencing of that gene in the P0 generation will prevent the transmission of transgenerational memory, which will result in the elimination of the lipid accumulation of the F1 generation. Alternatively, if a gene only functions as an executor, the silencing of that gene in the P0 generation should only influence the lipid level of P0 generation, so that we would still detect the elevation of fat level in F1 generation. Altogether, the results showed that silencing of *nhr-49* and *nhr-80* did not abrogate the lipid accumulation of F1 progeny (Fig. 2i, j), suggesting that *nhr-49* and *nhr-80* are purely executors but not transmitters; the silencing of *daf-16* or *sbp-1* in P0 generation caused the loss of lipid accumulation of F1 (Fig. 2h, k), indicating their roles as transmitters.

Taken together, our results demonstrated that *daf-16, nhr-49, nhr-80* and *sbp-1* were required for HFD-induced TEI of lipid accumulation. Among them, *nhr-49* and *nhr-80* functioned solely as executors; *sbp-1* was responsible for transmitting the heritable memories, though we could not rule out its role as an executor; for *daf-16*, it not only functions as an executor to regulate lipid accumulation but also as a transmitter to pass down heritable memory to progeny.

**Delta-9 desaturases mediated the heritable memories.** In *C. elegans*, delta-9 desaturases are well-characterized targets downstream of highly conserved transcription factors SBP-1, NHR-49, NHR-80 and DAF-16 (Fig. 3a)[23,31,35,36]. Because SBP-1, NHR-49, NHR-80 and DAF-16 have been detected to mediate TEI of lipid accumulation in our settings, we wondered whether delta-9 desaturases participated in the TEI of lipid accumulation. We used the delta-9 desaturases-related mutants including *fat-5, fat-6,* and *fat-7* lof mutants, and *fat-5; fat-6* and *fat-5; fat-7* double mutants to test our hypothesis. Our results showed that the fat accumulation phenotype remained largely unaffected in the P0 generation of these mutants; however, these mutations abrogated the elevated fat level in F1 and F2 progeny (Fig. 3b–f). These results revealed that delta-9 desaturases function in TEI induced by HFD. To more definitively specify the role of delta-9 desaturases in HFD-induced TEI, we conducted RNAi experiments similar to the above-mentioned treatments (*daf-16, nhr-49, nhr-80* and *sbp-1*). We found that silencing *fat-5* in F1 abolished the lipid accumulation (Fig. 3g), and silencing *fat-5, fat-6* or *fat-7* in P0 generation did not affect the phenotype of fat accumulation in their recovered F1 progeny (Fig. 3h and Supplementary Fig. 4f–h), suggesting that delta-9 desaturases solely execute regulation of lipid metabolism in HFD-induced TEI, but not transmit heritable memory.

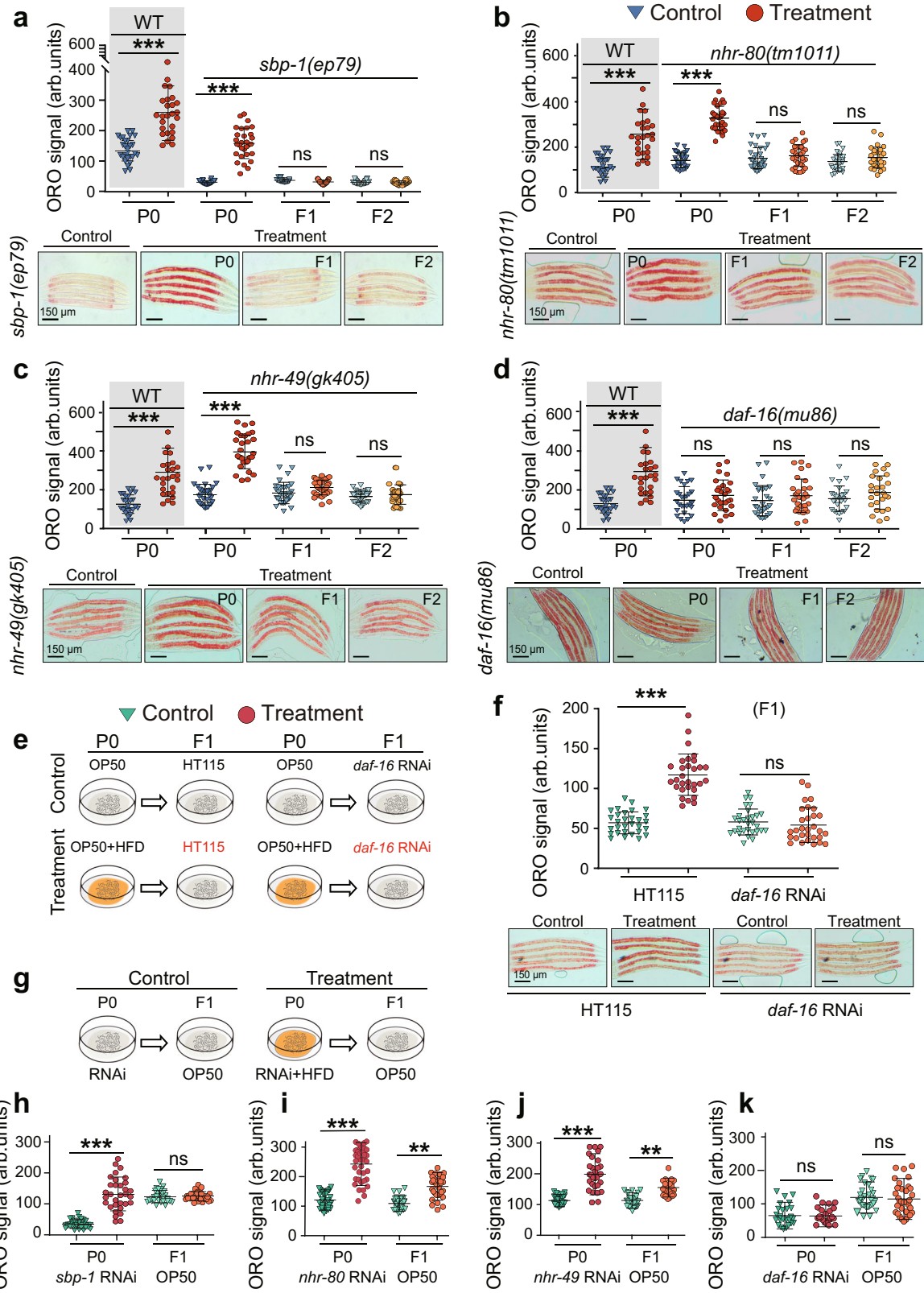

**Memory of HFD-induced lipid metabolism was conferred by histone H3K4me3 modification**. Reportedly, a set of chromatin-modifying enzymes, including deacetylases and H3K4me3 methyltransferase, influence lipid metabolism[27,37]. To understand whether methyltransferase mediated HFD-induced TEI of obesogenic effect, we performed a targeted mutant screening by selecting genes involved in histone methylation. We found that

loss of *wdr-5.1*, a H3K4me3 complex component, but not other H3K4me3 modifiers (i.e., *set-2* or *ash-2*), abolished the lipid accumulation in progeny (Fig. 4a and Supplementary Fig. 6a, b). Furthermore, *wdr-5.1* did not execute in responding to lipid metabolism induced by HFD, because ORO signaling of the *wdr-5.1* mutant remained mostly increased in the P0 parents (Fig. 4a). Moreover, we detected that the mRNA level of *wdr-5.1* was

**Fig. 2 NHR-49, NHR-80, SBP-1, and DAF-16 contribute to transgenerational inheritance of lipid accumulation.** Transgenerational inheritance test of lipid accumulation induced by HFD in *sbp-1* mutant ($n \geq 27$ per condition) (**a**), *nhr-80* mutant ($n \geq 29$ per condition) (**b**), *nhr-49* mutant ($n \geq 29$ per condition) (**c**), and *daf-16* mutant ($n \geq 29$ per condition) (**d**). For **a**–**d**, graph data are presented as mean ± SD (arb. units: arbitrary units); unpaired two-tailed Student's *t*-test; ***$p < 0.001$, ns: not significant. The same WT control was shown in **a**–**d** (for **a**–**d**, performing ORO staining analyses at the same time and using the same WT control). **e**, **f** P0 worms were fed with or without HFD and without any RNAi exposure, and then F1 progeny were exposed to RNAi with *daf-16* to test the requirement of *daf-16* in F1s. Quantification of ORO staining of F1 worms (mean ± SD; $n \geq 30$ per condition; unpaired two-tailed Student's *t*-test; ***$p < 0.001$, and ns: not significant). **g**–**k** DAF-16 and SBP-1, but not NHR-49 or NHR-80, contribute to transmit the transgenerational information of lipid accumulation. **g** Experimental scheme. **h**–**k** P0 worms were fed with or without HFD, and subjected to *sbp-1* RNAi ($n \geq 31$ per condition) (**h**), *nhr-80* RNAi ($n \geq 30$ per condition) (**i**), *nhr-49* RNAi ($n \geq 30$ per condition) (**j**) and *daf-16* RNAi ($n \geq 27$ per condition) (**k**), then F1 generation raised at OP50. Quantification of ORO staining of F1 worms (mean ± SD; unpaired two-tailed Student's *t*-test; ***$p < 0.001$, **$p < 0.01$, and ns: not significant). For **a**–**d** and **f**, scale bar = 150 μm. Source data are provided as a Source Data file.

slightly (but statistically significant) upregulated when animals were fed HFD compared with normal food (Supplementary Fig. 6g). By contrast, our results showed that other histone modifications including H3K27me3 and H3K36me3 did not contribute to the TEI induced by HFD, because lipid accumulation induced by HFD was not abrogated in the P0, F1 and F2 generations in the H3K27 demethylase lof mutant *jmjd-3.1*[38] and in the H3K36 methyltransferase lof mutant *met-1*[8] (Supplementary Fig. 6c, d). Furthermore, we observed that H3K4me3 methylation was significantly increased in the P0 generation fed with HFD (Fig. 4b) and in their recovered progeny (Fig. 4b–d and Supplementary Fig. 6e, f). Moreover, upregulation of H3K4me3 methylation induced by feeding with HFD was abolished in a *wdr-5.1* mutant (Supplementary Fig. 6i). These results suggested that histone H3K4me3 modification, and only histone H3K4me3 modification conferred by *wdr-5.1*, plays a specific role in transmitting the heritable memory induced by HFD.

To more clearly characterize the role of *wdr-5.1* in HFD-induced TEI, we also conducted two sets of RNAi experiments similar to the above-mentioned treatments (*daf-16, nhr-49, nhr-80* and *sbp-1*). We found that silencing *wdr-5.1* in F1 abrogated the lipid accumulation (Supplementary Fig. 7a), and silencing *wdr-5.1* in P0 generation also abolished the phenotype of fat accumulation in their recovered F1 progeny (Supplementary Fig. 7b), suggesting the role of *wdr-5.1* as a transmitter instead of an executor. Collectively, these results suggested that histone H3K4me3 modification plays a specific role in transmitting the heritable memory induced by HFD.

To test whether lipid accumulation induced by HFD is associated with heritable changes of H3K4me3 at specific loci, we conducted RNA sequencing (RNA-seq) and chromatin immunoprecipitation and sequencing (ChIP-seq) to compare changes in the enrichment of H3K4me3 peaks. The results exhibited that when animals fed with HFD, the transcription level and H3K4me3 level of many genes, especially lipid metabolism-associated genes (i.e., *daf-16, sbp-1* and *gmd-2*), was upregulated (Supplementary Fig. 7c–i). We used ChIP-qPCR and found that H3K4me3 occupancy was significantly increased at the promoter of these genes related to metabolism in the P0, F1 and F2 generations (Fig. 4f). The mRNA level of these genes was also significantly upregulated in the P0, F1 and F2 generations by analysis of RT-qPCR (Fig. 4g).

We further investigated whether the transcription factors (*sbp-1* and *daf-16)* and nuclear receptors (*nhr-49* and *nhr-80*), which have been identified to mediate HFD-induced TEI, also regulated the levels of histone H3K4me3 modification. We found that all mutations (*daf-16, nhr-49* and *nhr-80*) except *sbp-1* had no effect on H3K4me3 modification in animals fed with HFD when compared with controls (Fig. 4e and Supplementary Fig. 8a–d). In addition, the elevated transcription level of *wdr-5.1* induced by HFD was abrogated in the *sbp-1* mutant (Supplementary Fig. 6g). These findings suggested that *sbp-1* might mediate HFD-induced

memory transmission of lipid accumulation through upregulating *wdr-5.1* expression to increase the H3K4me3 level.

**Characterization of tissue-specific functions of *daf-16* and *sbp-1* in HFD-induced TEI.** To detect tissues in which *daf-16* and *sbp-1* function in response to lipid metabolism and in regulating histone modifications, we conducted tissue-specific gene knockdown assays by using strains that can process RNAi efficiently only in specific tissues, such as germline, muscle, or neuron (Fig. 5a). Firstly, we performed the tissue-specific RNAi of *daf-16* to detect the ORO staining. We found that muscle-specific RNAi of *daf-16* in the P0 generation led to the suppression of lipid accumulation (Fig. 5g). However, other tissue-specific RNAi, including in the intestine, neuron or germline, could not compromise the elevated fat level (Fig. 5h–k). It is interesting that DAF-16 plays a role in the muscle to regulate HFD-induced lipid accumulation which mainly occurs in the intestine. We wondered how DAF-16 senses the signal. Could it be the components of the insulin signaling pathway upstream of DAF-16 that play a role to transmit signals from the intestine to the muscle? To determine this hypothesis, we performed the lipid metabolism-associated germline or intestine-specific RNAi of *daf-2* (an insulin receptor-like gene) to detect the ORO staining. Our results exhibited that both germline and intestine-specific RNAi of *daf-2* did not abrogate lipid accumulation induced by HFD (Supplementary Fig. 9d–f), suggesting that DAF-16 senses other signals instead of the signals transmitted by *daf-2* to regulate lipid metabolism.

It is well known that DAF-16 executes a response to environmental stress by translocating to the nucleus[39]. Indeed, we found that DAF-16 displayed obvious aggregation in the nucleus, especially in muscle, in the P0 generation fed with HFD and in their recovery F1 progeny (Supplementary Fig. 9a). Moreover, we found that protein levels of DAF-16 were obviously elevated in P0 worms fed with HFD and their recovered F1 progeny (Supplementary Fig. 9b, c).

We then performed tissue-specific knockdown of *sbp-1* to assess the change of histone H3K4me3 induced by HFD. We found that intestine-specific, muscle-specific, germline-specific, or intestine- and germline-specific RNAi of *sbp-1* abolishes the elevated H3K4me3 modification levels (Fig. 5b–f). These results indicated that *sbp-1* functions in all detected tissues to regulate HFD-induced H3K4me3 modification. Similar to *daf-16*, we also conducted the tissue-specific RNAi of *sbp-1* to detect the ORO staining. Consistent with the results from *sbp-1* lof mutant, knockdown of *sbp-1* in different tissues could not abrogate lipid accumulation induced by HFD (Supplementary Fig. 10a–e). Collectively, tissue-specific RNAi studies demonstrated that *daf-16* and *sbp-1* function in one or more tissues, suggesting that signals from different tissues might coordinately regulate TEI of lipid accumulation induced by HFD.

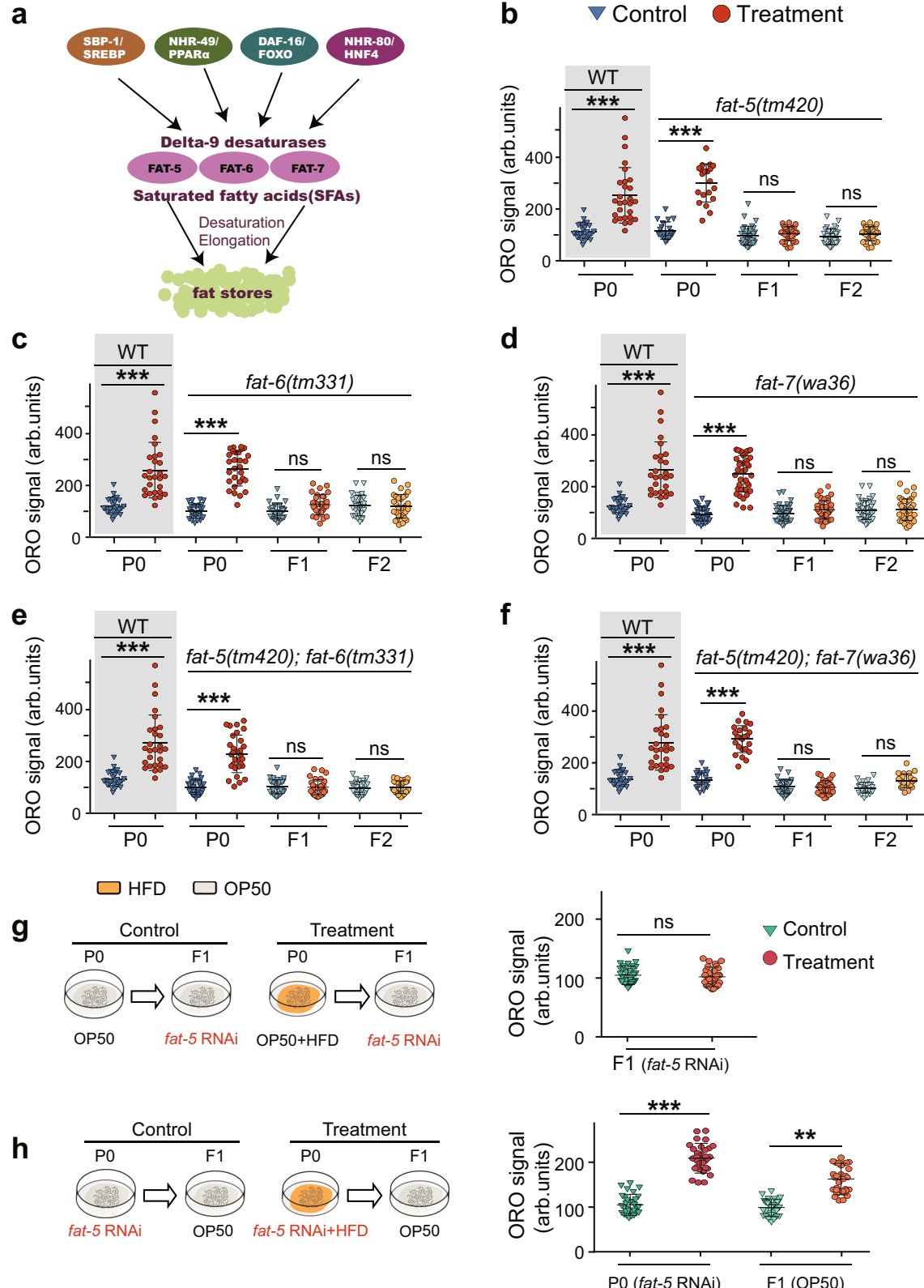

## Discussion

During the last several decades, the number of obese people has been growing rapidly in many countries, causing a huge economic burden to society[40]. However, for metabolic diseases such as obesity and T2D, besides genetic and environmental factors, family history has increasingly been considered to be an important factor. In fact, genetic background only contributes ~5–10%

of the overall risk for developing T2D[5]. Recently, considerable efforts have been done to elucidate the TEI-associated obesity induced by overnutrition. For example, studies in *Drosophila* have shown that obese male flies fed with a high-sugar diet can transgenerationally transmit lipid accumulation information through epigenetic markers (histone methylation modification) to their offspring and induce obesogenic phenotype[41]. Other studies

**Fig. 3 Transgenerational inheritance of lipid accumulation depends on *fat-5*, *fat-6* and *fat-7*. a** Schematic presentation of the transcriptional network that regulates delta-9 desaturase gene. **b–f** Transgenerational inheritance test of lipid accumulation induced by HFD in *fat-5* mutant ($n \geq 28$ per condition) (**b**), *fat-6* mutant ($n \geq 28$ per condition) (**c**), *fat-7* mutant ($n \geq 28$ per condition) (**d**), *fat-5; fat-6* double mutant ($n \geq 28$ per condition) (**e**), and *fat-5; fat-7* double mutant ($n \geq 30$ per condition) (**f**) (mean ± SD; unpaired two-tailed Student's *t*-test; ***$p < 0.001$, and ns: not significant). The same WT control was shown in **a–f**. **g** P0 worms were fed with or without HFD, and F1 progeny were exposed to *fat-5* RNAi to determine the requirement of *fat-5* in the F1 generation. **h** P0 animals were fed with or without HFD and subjected to *fat-5* RNAi, and then lipid levels of F1 progeny were analyzed. For **g** and **h** (right panels), graph data are presented as mean ± SD, $n \geq 31$ per condition, statistical analyses were performed by unpaired two-tailed Student's *t*-test; ***$p < 0.001$, **$p < 0.01$, and ns: not significant, Source data are provided as a Source Data file.

have shown that obese mice fed with HFD can transmit lipid accumulation information to their offspring and increase adiposity, which is conferred by changes in sperm miRNA and reduction in total global germ cell DNA methylation[42,43]. However, many mechanisms remain uncovered. In this work, we use *C. elegans* as an obesogenic model by feeding them HFD (egg yolk). Our results demonstrated that an obesity effect in parents in *C. elegans* could be induced by feeding egg yolk, and could be transmitted to their naive progeny that had never been exposed to HFD.

In addition, our study revealed that TEI of lipid accumulation was mediated by histone H3K4me3 modification. Consistent with previous reports[34,35], we found that DAF-16 is an essential regulator in response to change in lipid metabolism induced by HFD. Furthermore, we observed that *sbp-1*, *daf-16*, *nhr-49*, *nhr-80*, and delta-9 fatty acid desaturase genes (*fat-5*, *fat-6*, and *fat-7*) were responsible for TEI of obesogenic effects. Among them, *nhr-49* and *nhr-80* contributed to regulating lipid metabolism; *sbp-1* transmitted transgenerational memory; while *daf-16* acted in both regulation of lipid metabolism and transmission of obesogenic information. Strikingly, *sbp-1* itself also contributed to regulating the level of histone H3K4me3 modification. Consistent with our observations, a previous study has discovered that in *C. elegans*, a complex transcriptional network consisting of SBP-1, MDT-15, DAF-16, NHR-49, and NHR-80 regulates the activity of the delta-9 fatty acid desaturase genes (*fat-5*, *fat-6*, and *fat-7*). In turn, these factors affect the modification of H3K4me3, ultimately affecting the lifespan and lipid metabolism of the nematodes[27].

Taken together, these results suggest that, upon animals fed with HFD, the stress of lipid accumulation in the parental generation induces the activity of lipid metabolic transcription factors *sbp-1*, *daf-16*, *nhr-49*, and *nhr-80*. At the same time, *sbp-1* regulates histone H3K4me3 modification and establishes the epigenetic marks in descendants. In turn, the H3K4me3 marks in the progeny promote the recruitment of lipid metabolism-related genes (i.e., *sbp-1* and *daf-16*) and facilitate their activation; simultaneously, the activation of *daf-16* and *sbp-1* might recruit *nhr-49* or *nhr-80* to form complex, and then *daf-16*, *sbp-1*, *nhr-49* and *nhr-80* synergistically induce the expression of lipid metabolism-related genes (i.e., *fat-5*, *fat-6* and *fat-7*) to respond lipid metabolism and ultimately reset the metabolic processes, thereby completing the TEI of obesity effect (Fig. 6).

In this study, egg yolk was fed as HFD to induce an obesity phenotype in *C. elegans*. We found that, except for *daf-16*, loss of function of other genes, including *nhr-49*, *nhr-80*, and *sbp-1*, which are recognized as central regulators of lipid metabolism, could not completely block lipid accumulation induced by HFD in the P0 generation. A possible explanation is that the composition of the egg yolk is complex so that these genes only partially mediate the egg yolk-induced lipid accumulation. Because our work focused on the transgenerational inheritance of lipid accumulation, we have not yet fully elucidated the underlying details of molecular mechanisms that regulate lipid metabolism changes induced by egg yolk. This will be a subject of future work. Because egg yolk is used as HFD to induce the phenotype of lipid

aggregation, it is worth noting that the executors for lipid accumulation in the P0 and F1/F2 are different since they are fed with a different diet. Therefore, in our present study, for P0 generation, only *daf-16*, but not *nhr-49*, *nhr-80*, *sbp-1*, *fat-5*, *fat-6* and *fat-7* was necessary as an executor to respond to lipid metabolism induced by egg yolk; while, for F1 generation, all of them (including *daf-16*, *nhr-49*, *nhr-80*, *sbp-1*, *fat-5*, *fat-6* and *fat-7*) are required as executors to respond to lipid accumulation.

In addition, it is still unclear how *sbp-1* regulates the level of histone H3K4me3 modification in this work. Epigenetic mechanisms include chromatin structure and modification, DNA methylation and noncoding RNAs. In this study, we identified histone H3K4me3 modification as a contributing factor to the TEI of obesity effect; however, the function, if any, of DNA methylation and noncoding RNAs remains ambiguous. Therefore, more work is needed to further disentangle the molecular mechanisms of TEI related to obesity, including how and when the epigenetic memory is established, as well as how lipid metabolism genes interplay with genes involved in epigenetic inheritance. Our research provides evidence that the lifestyle of parents plays an important role in the metabolic status of offspring in nematodes, and it is tempting to speculate that these findings may be relevant to mammalian systems.

## Methods

**Nematode strains and maintenance.** *C. elegans* strains used in this work were obtained from *Caenorhabditis* Genetic Center (University of Minnesota, USA), which is supported by the NIH NCRR, and the National BioResource Project. All strains were maintained on standard NGM plates with *Escherichia coli* OP50 as previously described[44]. All experiments were performed at 20 °C. Strains used in this work were: wild type (N2), RB1304 *wdr-5.1(ok1417) III*, ZR2 *jmjd-3.1(gk384) X*, VC1666 *met-1(ok2172) I*, BX165 *nhr-80(tm1011) III*, BX107 *fat-5(tm420) V*, BX110 *fat-6(tm331) IV; fat-5(tm420) V*, BX106 *fat-6(tm331) IV*, BX160 *fat-7(wa36) fat-5(tm420) V*, BX153 *fat-7(wa36) V*, CE541 *sbp-1(ep79) III*, VC870 *nhr-49(gk405) I*, CF1038 *daf-16(mu86) I*, CF1553 muIs84 [(pAD76) *sod-3p*::GFP + rol-6(su1006)], NR350 kzIs20 [*hlh-1p*::rde-1 + sur-5p::NLS::GFP], AMJ345 *jamSi2* [*mex-5p*::rde-1(+)] II, TU3401 uIs69 [pCFJ90 (*myo-2p*::mCherry) + *unc-119p*::sid-1], MAH23 *rrf-1(pk1417) I*, VP303 kbIs7 [*nhx-2p*::rde-1 + rol-6(su1006)], STE70 *nhr-80(tm1011) III*, TJ356 zIs356 [*daf-16p*::daf-16a/b::GFP + rol-6(su1006)], VC974 *set-2(ok1484) III*, tm1726 *ash-2(tm1726) II*, MT18143 nIs286 [*mir-71*(+) + sur-5::GFP] X. New strains were built by standard genetic crosses and genotypes were confirmed by PCR or sequencing. Here, TJ356 was crossed with BX165 or VC870 to create double mutants *daf-16 (OE); nhr-80 (lof)* or *daf-16 (OE); nhr-49 (lof)*, respectively.

**Mating protocol.** Mating experiments were conducted with a GFP⁺ strain MT18143 to distinguish the cross progeny from the self-fertilized progeny. Briefly, synchronized L1 male (GFP⁺) or female (wild-type N2) worms were fed with HFD. Afterwards, worms (a ratio of males and females of 3:1) were picked onto OP50 seeded NGM plates to mate for 12 h at 20 °C. Two days after mating, GFP⁺ F1 young adult worms were picked and collected to perform ORO staining.

**RNA interference.** RNAi was essentially conducted using the standard feeding protocol. The *E. coli* HT115 transformed with vectors expressing the corresponding dsRNA and empty vector were derived from the Ahringer library (Source Bioscience, Nottingham, UK). All vectors were confirmed by sequencing. RNAi bacteria were cultured at 37 °C in LB with 100 µg mL⁻¹ ampicillin. Freshly prepared bacteria were spotted on NGM plates with 1 mM isopropyl-B-D-thiogalactoside before use. All RNAi treatments began from synchronized L1 larvae.

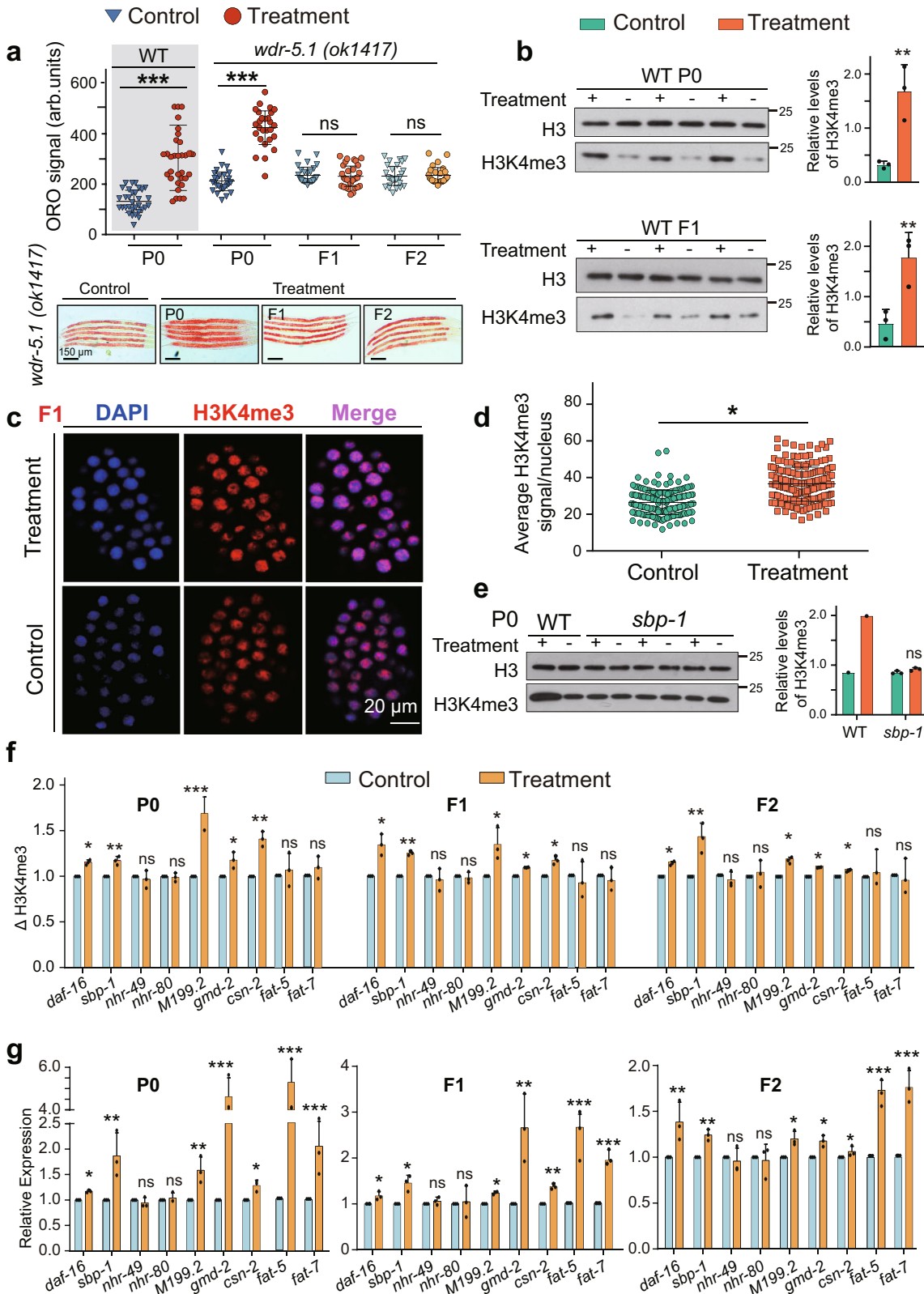

**Fatty acid supplementation**. All fatty acids and cholesterol used in this study were purchased from Sigma-Aldrich (Munich, Germany). For the fatty acid supplementation, a final concentration of 4 mM OA, 4 mM PA and 50 mg/mL cholesterol were used per experiment. OP50 were mixed with corresponding fatty acid and then seeded onto plates at room temperature for 6 h before the addition of worms. For egg yolk supplementation, the cooked random egg yolk was ground and mixed with OP50 (volume ratio, 1:5) to provide the *C. elegans* a food supplement. The mixture of egg yolk and bacteria is freshly made before each experiment.

**Conditions for inducing lipid accumulation**. To induce lipid accumulation, ~500 synchronized L1 larvae were raised on NGM plates with or without HFD. To obtain F1 generation worms, gravid day 1 adults treated with or without HFD were bleached and hatched overnight in 10 mL M9 buffer, and subsequently grown on NGM plates without HFD. This process was repeated to obtain the Fn generations. For worms fed with HFD until L4 larvae, synchronized L1 larvae were grown on NGM plates with HFD until L4 larvae, then transferred them to NGM plates with normal food, Fn generations animals were obtained using the similar protocol mentioned above.

**Fig. 4 Histone H3K4me3 modification mediates the transgenerational inheritance of lipid accumulation. a** Transgenerational inheritance test in *wdr-5.1* mutant (mean ± SD; $n \geq 28$ per condition; scale bar = 150 μm; unpaired two-tailed Student's *t*-test; ***$p < 0.001$, and ns: not significant). **b** Western blots (left panels) and quantification (right panels) of histone H3K4me3 modification in P0 animals fed with or without HFD, and their recovered F1 progeny (mean ± SD, $n = 3$ biologically independent samples, unpaired two-tailed Student's *t*-test, **$p < 0.01$). Immunostaining image (representative of three experiments) (**c**) and quantification (**d**) analyzed the level of histone H3K4me3 modification in F1 embryos from parents fed with or without HFD. Nuclei in blue, H3K4me3 in red. Mean ± SD from three independent experiments, each with $n = 10-20$ nuclei per worms, $n \geq 20$ worms per condition (mean ± SD, unpaired two-tailed Student's *t*-test, *$p < 0.05$). **e** The level of histone H3K4me3 modification in *sbp-1* mutant fed with or without HFD (mean ± SD; $n = 3$ biologically independent samples, unpaired two-tailed Student's *t*-test; ns, not significant). **f** H3K4me3 ChIP-qPCR of lipid accumulation response genes of wild-type animals fed with or without HFD and their recovery F1 and F2 progeny. **g** The mRNA levels of genes detected in (**f**). Mean ± SD; $n = 3$ biologically independent samples; unpaired two-tailed Student's *t*-test; ***$p < 0.001$, **$p < 0.01$ and *$p < 0.05$, ns: no significance. Source data are provided as a Source Data file.

**Oil-Red-O staining and quantification.** For ORO staining and quantification, all strains were cultured as described under conditions to accumulate lipid to obtain P0, F1 and F2 animals. Then animals were fixed in fixing solution (1× MRWB buffer (160 mM KCl, 40 mM NaCl, 14 mM Na$_2$EGTA, 1 mM spermidine-HCl, 0.4 mM spermine, 30 mM Na-PIPES, 0.2% β-mercaptoethanol) containing 1% paraformaldehyde) by gently rocking for 1 h at room temperature. Fixed worms were washed with 1×PBS, resuspended in 60% isopropanol and incubated for 15 min at room temperature. Worms were then incubated and stained using saturated ORO staining solution dissolved in 60% isopropanol overnight with rocking. Afterwards, worms were washed twice with 1× PBS 0.01% Triton X-100 to move excessive dye. Then animals were mounted to agar plates for imaging at ×200 magnification with a Nikon Ti2-U microscope. The same exposure setting was used across all conditions within each experiment. The ORO intensity (arbitrary unit, arb. units) was measured using ImageJ processing software as previously described[45] after background was removed and greyscale converted[45]. Statistical significance was analyzed using unpaired two-tailed Student's *t*-test by GraphPad Prism (v 8.0.1). At least 30 worms were used per experiment, and the experiment was repeated at least three times.

**Quantitative RT-PCR.** Briefly, about 4000 worms were used to extract total RNA with RNAiso Plus (Takara). Then RNA was reverse transcribed into cDNA using a cDNA Reverse Transcription Kit (ABclonal) as per the manufacturer's instructions. Real-time qPCR was conducted in triplicate for each gene using a CFX96 Real-Time PCR system (Bio-Rad) with SYBR Green select master mix (ABclonal). The quantification was computed by $2^{-\Delta\Delta Ct}$ method after being normalized to gene *cdc-42*. In order to rule out the possibility of *cdc-42* expression being affected by HFD, we analyzed the level of *cdc-42* using other reliable reference genes when animals fed with HFD compared with normal OP50. The primers are summarized in Supplementary Table 1.

**RNA-seq.** RNA-seq libraries were prepared and sequenced by Novogene Corporation (Beijing, China). Briefly, young adult synchronized worms were harvested with cold PBS buffer. Total RNA was extracted using RNAiso Plus (Takara). RNA integrity was assessed using the RNA Nano 6000 Assay Kit of the Bioanalyzer 2100 system (Agilent Technologies, CA, USA). A total amount of 1 μg RNA per sample was used as library and sequence. The sequence was performed by cBot Cluster Generation System using TruSeq PE Cluster Kit v3-cBot-HS (Illumia) according to the manufacturer's instructions. Differential expression analysis was performed using the DESeq2 R package (1.20.0). Genes with a greater than twofold change and false discovery rate <0.01 were defined as differentially expressed genes (DEG). Functional annotation of DEG was analyzed through various databases, including Gene Ontology (GO), Kyoto Encyclopedia of Gene and Genomes (KEGG) and EuKaryotic Ortologous Groups. HFD-induced DEG were assigned functional categories using the Database for Annotation, Visualization and Integrated Discovery.

**H3K4me3 ChIP-qPCR.** Embryos were collected and flash-frozen in liquid nitrogen. Embryo pellets (50–100 μL) were incubated with ChIP cross-linking buffer (M9 containing 2% formaldehyde (Sigma)) at room temperature for 30 min. After quenching with glycine (125 mM final), the pellets were washed three times in PBS. Afterwards, samples were resuspended in FA buffer (50 mM HEPES/KOH [pH 7.5], 1 mM EDTA, 1% Triton X-100, 0.1% sodium deoxycholate, and 150 mM NaCl, 1% proteinase inhibitor cocktail (Roche)) and sonicated with a Bioruptor at maximum power for 30 s on and 30 s off. Samples were then centrifuged at 15,000 ×g for 10 min at 4 °C, and the supernatants were stored at –80 °C. For H3K4me3 ChIP experiments, chromatin extract (1 mg of total protein) was thawed, pre-cleared, then immunoprecipitated with 5 μL H3K4me3 antibody (Millipore 04-745) at 4 °C for 12–16 h. Next, 40 μL pre-blocked SureBeads$^{TM}$ Starter Kit Protein G (Bio-Rad) were added, and the tubes were rotated at 4 °C for 4 h. Beads were washed with 1 mL of the following buffers for 5 min at room temperature: twice with FA buffer, once with FA-1M NaCl buffer (50 mM HEPES/KOH [pH 7.5], 1 mM EDTA, 1% Triton X-100, 0.1% sodium deoxycholate, and 1 M NaCl), once

with FA-500 mM NaCl buffer (50 mM HEPES/KOH [pH 7.5], 1 mM EDTA, 1% Triton X-100, 0.1% sodium deoxycholate, and 500 mM NaCl), twice with TEL buffer (0.25 M LiCl, 1% NP-40, 1% sodium deoxycholate, 1 mM EDTA, 10 mM Tris-HCl (PH 8.0)), and finally with twice 1×TE buffer. Samples were eluted using elution buffer (1% SDS in TE with 250 mM NaCl). Supernatants and input samples were incubated with proteinase K (0.1 μg/mL) for 2 h at 50 °C and then de-crosslinked overnight at 65 °C. DNA was purified before performing qPCR with SYBR. The quantification was computed by the $2^{-\Delta\Delta Ct}$ method. The primer from an intergenic region on chromosome IV was used as an internal control, as previously described[46]. Sequences of qPCR primers are listed in Supplementary Table 1.

**H3K4me3 ChIP-seq.** ChIP-seq libraries were prepared and sequenced by Novogene Corporation (Beijing, China). Briefly, 3 g of *C. elegans* were harvested and washed with cold PBS buffer, then crosslinked with 1% formaldehyde in M9 buffer. After lysing the sample to obtain chromatins, chromatins were sonicated to get soluble sheared chromatin (average DNA length of 200–500 bp). The soluble chromatin was used for immunoprecipitation by H3K4me3 antibody (5 μL per sample) (Abcam ab8580) and 5% of it was saved as input. DNA purity was checked using the NanoPhotometer® spectrophotometer (IMPLEN, CA, USA). The purified DNA was used for ChIP-seq library preparation and sequence on Illumina platform (Illumina, CA, USA). Library quality was assessed on the Agilent Bioanalyzer 2100 system. Raw data (raw reads) of fastq format were firstly processed using fastp (v 0.19.11) software[47]. Clean data (clean reads) were obtained by removing reads containing adapter and ploy-N, as well as low-quality reads. Index of the reference genome was built using BWA (v 0.7.12) and clean reads were aligned to the reference genome using BWA mem (v 0.7.12). Afterwards, regions of IP enrichment over background were identified by the MACS2 (v 2.1.0) peak calling software (q-value threshold of 0.05 was used for all data sets). Peak-related genes were confirmed by ChIPseeker (v 3.14). And then GO enrichment analysis was performed by the GOseq R package. KEGG analysis was conducted by KOBAS software. The heatmaps, reads distribution of diffpeak, and all peaks were obtained using Deeptool (v 3.2.1).

**Western blot analysis.** Young adult worms treated as described under conditions to accumulate lipid were collected with M9 buffer, washed three times and pellets were snap-frozen in liquid nitrogen and stored at –80 °C. Pellets were lysed in RIPA buffer. Next, pellets were ground twice using a TissueLyser at 75 Hz for 6 min at 4 °C, and centrifuged at 10,000 ×g at 4 °C. Supernatants were collected. All supernatants were quantified with a BCA Protein Assay Kit. Worm RIPA samples were boiled at 95 °C for 5 min before being resolved on SDS-PAGE (13.5%) and transferred to nitrocellulose membrane. The membranes were blocked in 5% milk, then incubated with primary antibodies to H3K4me3 (1:3000, Millipore 04-745), H3 antibody (1:10,000, CST, H9715), GFP (1:5000, ROCHE, 11814460001) or actin (1:5000, Sigma, A1978). The primary antibody was visualized using horseradish peroxidase-conjugated anti-rabbit secondary antibody (1:5000, Jackson ImmunoResearch, 111-035-144) and ECL Western Blotting Substrate.

**Worm eggs immunocytochemistry.** For immunostaining of worm eggs, eggs were bleached from gravid worms and washed three times with M9 buffer. Eggs were resuspended in fixing solution (160 mM KCl, 40 mM NaCl, 20 mM Na2EGTA, 10 mM spermidine-HCl, 30 mM Na-PIPES, 50% methanol, 2% beta-mercaptoethanol, 2.5% polyformaldehyde) and frozen in liquid nitrogen for 10 min. Fixed eggs were kept at –80 °C for long term storage. Before staining, eggs were thawed and fixed at 4 °C for 30 min, and washed twice using Tris-Triton buffer (100 mM Tris-HCl pH 7.4, 1 mM EDTA, 1% Triton X-100) for 5 min. Eggs were then blocked with PBST-A buffer (PBS PH 7.4, 1% BSA, 0.5% Triton X-100, 5 mM sodium azide, 1 mM EDTA) for 20 min and incubated overnight with primary antibodies to H3K4me3 (1:100 in PBST, Millipore 04-745). Eggs then were washed three times, each time for 10 min, with PBST-B (PBS PH 7.4, 0.1% BSA, 0.5% Triton X-100, 5 mM sodium azide, 1 mM EDTA), and then incubated with Alexa Fluoro 546 secondary antibody (1:300). DAPI (2 mg mL$^{-1}$) was added to visualize

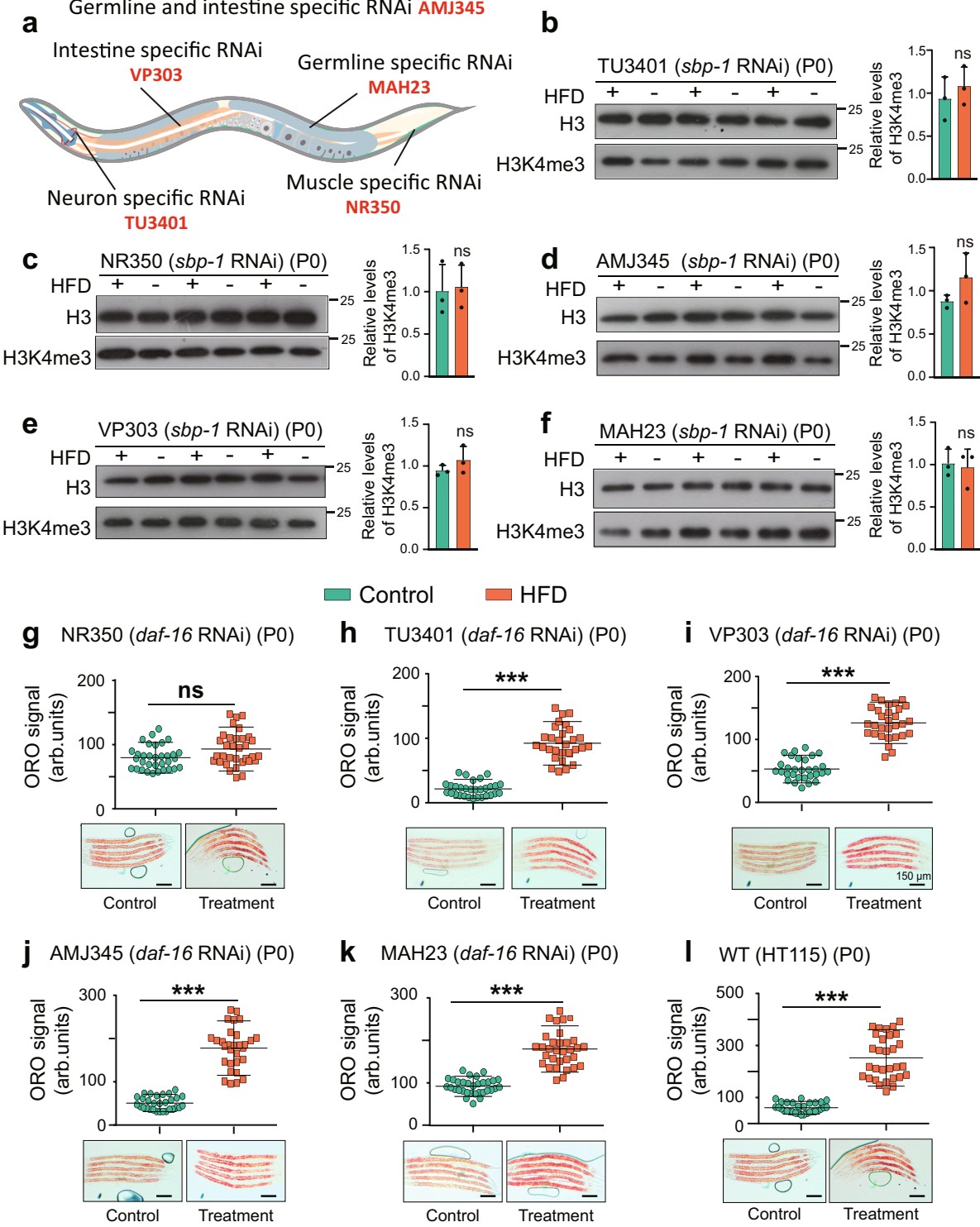

**Fig. 5 Tissue-specific RNAi of *daf-16* or *sbp-1*. a** Strains used for the tissue-specific gene knockdown. **b**–**f** Western blots (left panels) and quantification (right panels) of histone H3K4me3 modification in worms subjected to neuron-specific *sbp-1* RNAi (**b**), muscle-specific *sbp-1* RNAi (**c**), germline and intestine-specific *sbp-1* RNAi (**d**), intestine-specific *sbp-1* RNAi (**e**), germline-specific *sbp-1* RNAi (**f**) (mean ± SD; *n* = 3 biologically independent samples; unpaired two-tailed Student's *t*-test; ns, not significant). **g**–**l** quantification of ORO staining in animals subjected to muscle-specific *daf-16* RNAi (*n* ≥ 31 per condition) (**g**), neuron-specific *daf-16* RNAi (*n* ≥ 30 per condition) (**h**), intestine-specific *daf-16* RNAi (*n* ≥ 30 per condition) (**i**), germline and intestine-specific *daf-16* RNAi (*n* ≥ 30 per condition) (**j**), germline-specific *daf-16* RNAi (*n* ≥ 31 per condition) (**k**) and without any RNAi exposure (*n* ≥ 30 per condition) (**l**). For **g**–**l**, graph data are presented mean ± SD; unpaired two-tailed Student's *t*-test; ***$p < 0.001$, ns: no significance; scale bar = 150 μm. Source data are provided as a Source Data file.

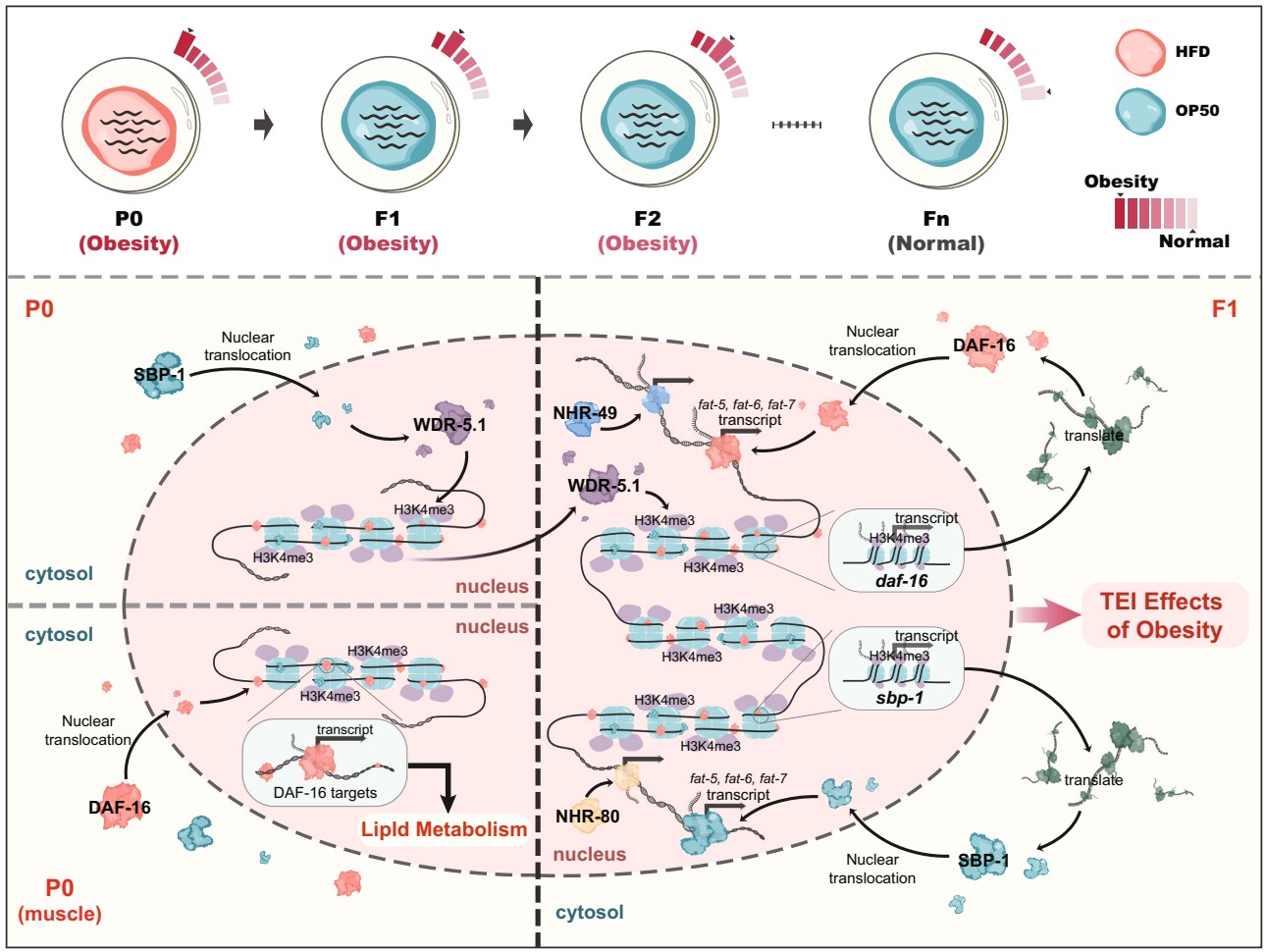

**Fig. 6 Overview model of transgenerational inheritance in response to lipid accumulation induced by the HFD.** HFD can induce a TEI phenotype of lipid accumulation. In worms fed with HFD, the stress of lipid accumulation in the parent promotes the activity of *sbp-1*, *daf-16*, *nhr-49*, and *nhr-80*. Among them, only *sbp-1* is involved in regulating histone H3K4me3 modification in a *wdr-5.1*-dependent manner and establishing the epigenetic marks in descendants. The upregulation of H3K4me3 modification promotes the recruitment and activation of *daf-16* and *sbp-1*; simultaneously, the activation of *daf-16* and *sbp-1* might recruit *nhr-49* or *nhr-80*, which then synergistically induce the expression of *fat-5*, *fat-6* and *fat-7* to respond lipid metabolism, thereby completing the TEI of obesity effect.

nuclei. Eggs were mounted to a microscope slide and visualized using a Zeiss Axio Imager Z2 with Apotome.2 microscope at 630-fold magnification. The fluorescence intensity was analyzed using ImageJ software. Statistical significance was analyzed using unpaired two-tailed Student's *t*-test. At least 20 eggs and 10–20 nuclei per egg were used per experiment, and experiments were repeated at least three times.

**Fluorescence microscopic imaging**. For quantification of SOD-3, synchronized L1 larvae of CF1553 (SOD-3p::GFP) were treated as described above to induce accumulation of lipid. Young adult worms were mounted to agar plates for imaging using a Nikon Ti2-U fluorescence microscope with ×20 air objectives. To exhibit the fluorescence intensity, six to seven randomly selected worms were put together to take images. For quantification purpose, each worm was imaged individually. The GFP fluorescence intensity was quantified by mean fluorescence intensity (total intensity/area) using ImageJ software. At least 30 animals were used per experiment. Statistical analyses were performed using unpaired two-tailed Student's *t*-test.

**Lifespan assay**. Lifespan assays were performed at 20 °C, based on standard protocols, as previously described[17]. In brief, ~100 young adults fed with HFD from L1 stage were transferred to NGM plates with 10 μM 5-fluoro-2'-deoxyuridine (Sigma) seeded with heat-inactivated OP50 to perform survival analyses. This day was defined as day 1. Death events were scored daily. The experiments were repeated at least twice. The mean, SEM, *p* and lifespan values were processed using the Kaplan–Meir survival analysis in SPSS software (v 26).

**Reporting summary**. Further information on research design is available in the Nature Research Reporting Summary linked to this article.

## Data availability

The data that support this study are available from the corresponding author upon reasonable request. The RNA-seq and ChIP-seq data generated in this study have been deposited in the Sequence Read Archive (SRA) database under accession code PRJNA770182 and PRJNA770442. Source Data are provided with this paper.

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

## Acknowledgements

We would like to thank Dr Xuhui Lai from Abclonal for technical help on our qPCR and RNA-seq procedures. We would like to thank Dr Huai-Rong Luo for providing strains, protocols and suggestions. We appreciate Dr David Katz's constructive comments when he reviewed our manuscript that was considered (but eventually rejected) by another journal. This work was supported by the National Key R&D Program of China (2021YFA0804900) (Q.Z.), the National Natural Science Foundation of China (No. 82001465) (Q.-L.W.), the Program of Introducing Talents of Discipline to Universities (111 Project, No. B16021) (Q.Z.), the Science and Technology Plan Project of Guangzhou, China (202002030021) (Q.Z.), the Guangdong Provincial Basic Research Program, China (2020A1515111026) (Q.-L.W.). We would like to thank the *Caenorhabditis* Genetic Center (CGC) for providing the worm strains, which is funded by the NIH Office of Research Infrastructure Programs (P40OD010440). Q.Z. also gratefully acknowledge the support of K.C. Wong Education Foundation.

## Author contributions

Q.Z. and Q.-L.W. designed the study. Q.-L.W., X.M., C.W., W.D., Z.L., J.Y., X.F. and Q.Y. conducted the experiments; Q.Z., Q.-L.W. and X.M. analyzed the data; Q.-L.W. wrote the manuscript; Q.Z., Z.J., Z.Y. and Z-H.Z. reviewed and edited manuscript; all authors commented on the manuscript.

## Competing interests

The authors declare no competing interests.
