## [Peer Review File · Nature Communications]

REVIEWER COMMENTS

Reviewer #1 (Remarks to the Author):

In the manuscript entitled "Histone H3K4me3 modification is a transgenerational epigenetic signal for lipid metabolism in *Caenorhabditis elegans*", Wan et al. investigate the mechanism of transgenerational inheritance of obesity predisposition from parents fed a high fat diet. In mammals, obesity and metabolic disorders can be epigenetically pre-programmed in progeny and grand-progeny of individuals that experienced either under or over nutrition during critical periods in development. The mechanisms of how the "memory" of nutritional status is passed through the germ line to metabolically program the next generation is not well understood. In this manuscript, the authors establish *C. elegans* as a model for the transgenerational epigenetic inheritance (TEI) of over-nutrition states. Hermaphrodites fed a high fat diet during development have progeny and grand-progeny that exhibit increased lipid storage compared to control populations fed a standard *E. coli* diet. The authors also show that the TEI phenotype of increased lipid storage is eliminated in strains carrying mutations in genes with functions in metabolism. In addition, they identify WDR-5.1, which is a member of the COMPASS complex, as also playing a role in TEI of increased lipid storage. Histone H3K4me3 modifications are increased in animals fed a high fat diet, as well as their F1 and F2 progeny, suggesting a link between the histone modification and inheritance of metabolic states. This manuscript has the potential to be of value to the metabolism research field; however, some of the conclusions are not well supported and the connection between H3K4me3 and lipid storage levels is not clear.

Major concerns:

1. The manuscript contains problems with grammar and typos, some of which make the results difficult to decipher. For example, on page 4, "machinery that has until now been elucidated". In some instances, "either/or" is used instead of "and", such as on page 5, "We found that the ORO levels were significantly increased, in either the HFD-fed parents or their F1 and F2 progeny (fed with normal OP50)".
2. This TEI phenotype is so clear and convincing in wild type that I am surprised that it has not been described previously given the amount of metabolism research in *C. elegans*. I noticed, however, that animal populations in these assays were synchronized using L1 diapause, which has transgenerational effects on gene expression (Rechavi and Baugh labs). L1 diapause is a starvation state, and DAF-16 is required for survival in that stage (Baugh lab). These facts make me question whether the increased lipid phenotypes are mechanistically connected to their L1 diapause experience. The authors should verify in wild type that their phenotype is reproducible in populations that have not experienced diapause or describe their results within that context.
3. The premise of this paper is that predisposition of obesity can be inherited. In the introduction, the authors give the example of the Dutch hunger winter in which children born to malnourished mothers during a famine had a higher probability of obesity and metabolic disorders in adulthood, which was passed on for at least two generations. However, the experimental model of this manuscript is that induced obesity from a high fat diet can predispose progeny to also be obese. I think the introduction should focus more on literature exploring this premise as an epigenetic phenomenon and what is known in mammals. Further, the discussion needs to be rewritten to put the results presented in greater context of metabolism research in *C. elegans* and obesity TEI research in other organisms.
4. Figure 1f and 1g: if the authors are examining for dosage effects that occur between populations with varying numbers of generations receiving retreatments, an ANOVA should be performed to compare all the treatment groups. Just comparing a single treatment to controls will not detect these effects.
5. I am not convinced about the role of SBP-1 in the TEI of lipid metabolism. For one, *sbp-1* mutants appear to have almost no stored lipids according to the ORO staining results compared to wild type, similar to what has been observed in other studies. Feeding *sbp-1* a high fat diet appears to rescue their lipid levels to wild type levels. A rescue of *sbp-1* stored lipid levels has previously been shown for

oleic acid supplementation (Yang et al Nature 2006). How do you eliminate the possibility that these mutants are unable to store lipids from the possibility that they are regulating TEI phenotype? This question is also relevant for Figure 3 examining the fat mutants. The possibility exists that *sbp-1* and the fat mutants have "received" the signal for altered lipid storage, but are not capable due to the nature of their mutation. I do not think it is surprising fat storage mutants do not exhibit increased lipid storage phenotypes.

6. P10: The authors examine the potential role of chromatin modifiers in the regulation of their TEI phenotype. They cite a review and Han et al. to justify the link between the H3K4me3 chromatin modification and lipid metabolism. However, they examine the mutants *met-1* (a H3K36me3 methyltransferase) and *jmjd-3.1* (a H3K27me2/3 demethylase) which is not justified. In Han et al. they show that members of the COMPASS complex, which includes the SET-2 H3K4me3 methyltransferase, regulates lipid metabolism genes, and that other complexes do not appear to regulate these pathways. The authors show convincing data that WDR-5.1 plays a role in TEI of increased lipid storage. This section would be strengthened by also examining *set-2* and *ash-2* in their assay.

7. The correlation between H3K4me3 levels and TEI of lipid storage is not clear. First, H3K4me3 appears elevated in *sbp-1* mutants in control and untreated samples compared to wild type, but do not show increased lipid storage (see comment above). This suggests that elevated H3K4me3 alone is not sufficient for this phenotype. Related to this, *fat-5* and *fat-7* genes are upregulated in their treatment animals (P0, F1, and F2), but H3K4me3 levels in their promoters are unchanged. Additionally, *NHR-49* and *NHR-80* are required for TEI, but these genes have no changes in gene expression or H3K4me3 levels. The authors should address these nuances in their data and explain a potential model for how H3K4me3 and lipid storage in more detail.

8. P6: very little attention is given to the RNA-seq data. As is, it adds little value to the results. What are the 25 genes that overlap with P0, F1, and F2 populations? In referring to these results, the manuscript states they are "consistent with a previous study", but there is no citation. In addition, the methods should describe in detail how the RNA-seq libraries were analyzed.

9. Lipid metabolism and chromatin modifications are mechanistically linked to lifespan. In general, adults with increased lipid storage and decreased H3K4me3 levels have increased lifespans. Since treatment animals in this study exhibit increased lipid storage AND increased H3K4me3 levels, the lifespan prediction is unclear. The authors have an opportunity to tease apart these two pathways in their regulation of an important life history trait.

Minor concerns:

P3: "In modern societies, people no longer live in fear of famine. Instead, we live in a condition of over-nutrition...". This statement is grossly incorrect. Many people around the world do not have food security. This sentence should be deleted.

P6: "One concern is that HFD-induced fat accumulation could trigger a maternally heritable response in a dose-dependent manner " Do the authors mean that the fat accumulated in the P0 is transmitted into F1 embryos via vitellogenesis, and that the extra lipids are maintained until adulthood and transmitted into the F2 embryos? This hypothesis seems highly unlikely, but should have a citation if it has been reported previously. I would recommend rewriting this part.

P11: "Germline to Soma Communication" is not demonstrated by Figure 5.

P10: "HFD-induced lipid accumulation in parents could just pass down to F2 generation, and F3 is back to normal lipid level, suggesting that HFD might induce epigenetic changes rather than genetic mutations in descendants" This is an odd statement to make towards the end of the results given how this phenotype has been labeled TEI from the beginning of the results.

Figure 2: There are differences in wild type ORO staining in the treatment groups that are not explained.

Reviewer #2 (Remarks to the Author):

In this study, Wan et al. showed that high fat diet induced lipid accumulation can pass to the next generations even they were fed with regular diet. They found that the transgenerational fat accumulation is regulated by transcription factors (DAF-16/FOXO and SBP-1), nuclear receptors (NHR-49 and NHR-80), and delta-9 desaturases. Furthermore, they demonstrated that the H3K4 trimethylation is enriched in the lipid metabolic genes, which promotes the multigeneration obesogenic effects. This study is novel and would be interesting to others in the community and the wider field. The data is convincing. Having said that, some questions need further address. First, the author performed RNAi and identified a list of genes up regulated by HFD. However, they did not go further to test them to see if any of them are required for transgenerational fat accumulation. Second, they showed many genes are required for the transgenerational fat accumulation, but the relationship among those genes has not been determined. For example, which genes initiate the signaling, how this signaling is passed to the others. From their data, it seems that DAF-16 acts upstream. But it did not affect H3K4me3, which suggests it is not required for the epigenetic changes required for the transgeneration effect. Could the author explain how DAF-16 acts as a "transmitter"?

Specific questions

1. RNA seq data should be validated, either by quantitative rt-PCR or examining if they are involved in the fat accumulation in P0 or F1/F2.
2. In the figure 2, the authors showed *nhr-49*, *nhr-80*, *sbp-1* and *daf-16* are required for the transgeneration fat accumulation. However, the relationship among them is not clear. They should at least perform some epistasis analysis. For example, they could overexpress *sbp-1/nhr-49/nhr-80* in the *daf-16* mutant to see if it could suppress the *daf-16* phenotype, or overexpressing *daf-16* in other mutant background. Alternatively, they can also examine if the expression of those factors is regulated by others.
3. The conclusion that *nhr-49* and *nhr-80* are purely executors based on the RNAi in P0. But it is also possible that the RNAi efficiency is low. They should exclude this possibility.
4. It is really interesting that they found DAF-16 is only required in muscle. But how does *daf-16* sense the HFD? Is insulin or *daf-2* involved?
5. I expect that the executors for lipid accumulation in the P0 and F1/F2 would be different since they are fed with different diet. The authors should discuss this further.
6. For data analysis, the number of replicates, the total number of animals should be indicated in the methods or figures. The authors normalized their data with *cdc-42*. Just curious, did they verified if *cdc-42* expression is affected by HFD? The immunoblot data should be quantified and analyzed. The data in the table S1 is not exactly correlated to the figure. For example, the figure 1a has five type of HFD treatment. Only one is showed in the table. And the description is not consistent. For example, in the figure, it shows P0-F5. In the table, it describes as P1, P4.
7. Other interesting questions including it would be interesting to know, if the fat accumulation signaling is maternal, paternal or both. They have showed feeding HFD from birth to L4 is sufficient to induce the transgeneration effect, but what is the minimum duration and the time window for feeding with HFD to induce the transgeneration effect?

Reviewer #3 (Remarks to the Author):

This manuscript started from an interesting phenotype that the high-fat diet (HFD) induces transgenerational lipid accumulation in *C. elegans*. The authors characterized a series of transcription factors involved in lipid metabolic regulation, including NHR-49, NHR-80, SBP-1, and DAF-16, and found that SBP-1 is more specifically responsible for transmitting effect between generations (transmitter), NHR-80 and NHR-49 mediate the lipid accumulation in the descendants (executor), while DAF-16 is not only required for the HFD-induced lipid accumulation, but also act as both an executor and transmitter. They further examined delta-9 desaturases, including fat-5, fat-6 and fat-7, and showed that fat-5 is an executor and mediates the lipid accumulation in the descendants. Furthermore, the authors found that HFD induces H3K4me3 in a sbp-1 dependent manner, and wdr-5.1 is specifically required for HFD-induced H3K4me3 and transmitting the effect between generations. Overall, the manuscript is well-written and easy to follow. The discovery will be interesting to a broad audience. To fully support the model, there are several major points that are required to be addressed.

Major points:

1. To distinguish an executor and a transmitter, the authors have designed experiments where RNAi feeding is applied in P0 but not F1 and the lipid phenotype is examined in F1. It is known that in some cases, RNAi knockdown can be passed from P0 to F1. This transgenerational RNAi effect can become a possible confounding factor for their experimental design. Given this effect varies among different RNAi clones, the authors should test whether gene knockdown occurs in F1 for all the RNAi clones.
2. The authors claimed that SBP-1 and DAF-16 can both function as transmitters. However, as shown in Figure 2h and k, the lipid accumulation in treated F1 is not suppressed with either sbp-1 or daf-16 P0 RNAi. Instead, the lipid level is increased in untreated F1. So the current results cannot support the claim.
3. Given the transgenerational effect does not pass to F3, the level of H3K4me3 should be also examined in F2 and F3. Based on the current model, the induction of H3K4me3 should be reduced over generations.
4. Inactivation of wdr-5.1 decreases the H3K4me3 level, which is the opposite to the HFD treatment. However, the wdr-5.1 mutant has increased lipid accumulation. Whether H3K4me3 also targets genes that trigger lipid catabolism? Are any of these genes found in the ChIP-seq data?
5. Comparison of RNA-seq and ChIP-seq data should be conducted to provide a systemic view on the the transcriptional landscape.
6. To further strengthen the connection between SBP-1, H3K4me3 and the lipid phenotype, tissue-specific knockdown of sbp-1 should be examined for the lipid phenotype in addition to the H3K4me3 level. Given that similar studies have been done for daf-16 (fig. 5g-i), there should not be technical difficulties.
7. Although inactivation of all delta-9 desaturases suppresses lipid accumulation in F1 and F2, only fat-5 is tested using the executor/transmitter distinguishing experiments. To support the claim that delta-9 desaturases are executors, the authors should also test fat-6 and fat-7.
8. Given that DAF-16 is both an executor and transmitter, does daf-16 knockdown affect H3K4me3 levels? Also, is Daf-16 nuclear translocation induced by HFD and transmitted to descendants? If yes, whether DAF-16 nuclear translocation specifically occurs in the muscle?

Minor points:

1. Figure 2h: the scale of the y axis is not correct.

Reviewer #1 (Remarks to the Author):

In the manuscript entitled “Histone H3K4me3 modification is a transgenerational epigenetic signal for lipid metabolism in Caenorhabditis elegans”, Wan et al. investigate the mechanism of transgenerational inheritance of obesity predisposition from parents fed a high fat diet. In mammals, obesity and metabolic disorders can be epigenetically pre-programmed in progeny and grand-progeny of individuals that experienced either under or over nutrition during critical periods in development. The mechanisms of how the “memory” of nutritional status is passed through the germ line to metabolically program the next generation is not well understood. In this manuscript, the authors establish C. elegans as a model for the transgenerational epigenetic inheritance (TEI) of over-nutrition states. Hermaphrodites fed a high fat diet during development have progeny and grand-progeny that exhibit increased lipid storage compared to control populations fed a standard E. coli diet. The authors also show that the TEI phenotype of increased lipid storage is eliminated in strains carrying mutations in genes with functions in metabolism. In addition, they identify WDR-5.1, which is a member of the COMPASS complex, as also playing a role in TEI of increased lipid storage. Histone H3K4me3 modifications are increased in animals fed a high fat diet, as well as their F1 and F2 progeny, suggesting a link between the histone modification and inheritance of metabolic states. This manuscript has the potential to be of value to the metabolism research field; however, some of the conclusions are not well supported and the connection between H3K4me3 and lipid storage levels is not clear.

Our response: Thank the reviewer’s positive feedback.

Major concerns:

- 1. The manuscript contains problems with grammar and typos, some of which make the results difficult to decipher. For example, on page 4, “machinery that has until now been elucidated”. In some instances, “either/or” is used instead of “and”, such as on page 5, “We found that the ORO levels were significantly increased, in either the HFD-fed parents or their F1 and F2 progeny (fed with normal OP50)”.*

Our response: We thank the reviewer for this comment. We revised these sentence as suggested and proofread the manuscript thoroughly.

- 2. This TEI phenotype is so clear and convincing in wild type that I am surprised that it has not been described previously given the amount of metabolism research in C. elegans. I noticed, however, that animal populations in these assays were synchronized using L1 diapause, which*

has transgenerational effects on gene expression (Rechavi and Baugh labs). L1 diapause is a starvation state, and DAF-16 is required for survival in that stage (Baugh lab). These facts make me question whether the increased lipid phenotypes are mechanistically connected to their L1 diapause experience. The authors should verify in wild type that their phenotype is reproducible in populations that have not experienced diapause or describe their results within that context.

Our response: We thank the reviewer for raising this point. To exclude the effect of starvation from synchronization in our setting. We conducted corresponding experiment in which mothers fed with HFD directly laid eggs and hatched to obtain their offspring without synchronization. Our results showed that transgenerational obesity effect in wild type is consistent in populations with or without starvation from synchronization, indicating that synchronization did not affect TEI in our settings. The corresponding results are shown in Fig. 1h (the following graph) in revised manuscript.

3 The premise of this paper is that predisposition of obesity can be inherited. In the introduction, the authors give the example of the Dutch hunger winter in which children born to malnourished mothers during a famine had a higher probability of obesity and metabolic disorders in adulthood, which was passed on for at least two generations. However, the experimental model of this manuscript is that induced obesity from a high fat diet can predispose progeny to also be obese. I think the introduction should focus more on literature exploring this premise as an epigenetic phenomenon and what is known in mammals. Further, the discussion needs to be rewritten to put the results presented in greater context of metabolism research in C. elegans and obesity TEI research in other organisms.

Our response: We thank the reviewer's suggestion. We have deleted the example about

“Dutch hunger winter.”. We have revised the manuscript to include additional discussion about obesity TEI research in *C. elegans* (Page 4, Line 5-8, and Page 13, Line 20-25), where we say:

“For example, it was reported that the TEI obesogenic effects of sulfomethoxazole are associated with histone H3K4me3 modification ¹⁵; another study indicated that a reduction in lipid accumulation induced by benzyloquinoline can be transmitted to progeny ¹⁶.”

“For example, study in *Drosophila* have shown that obese male flies fed with high-sugar diet can transgenerationally transmit lipid accumulation information through epigenetic markers (histone methylation modification) to their offspring and induce obesogenic phenotype ³⁷. Other studies have shown that obese mice fed with HFD can transmit lipid accumulation information to their offspring and increase adiposity, which is conferred by changes in sperm miRNA and reduction in total global germ cell DNA methylation ^{38,39}.”

Reference:

R15. Li, Z., Yu, Z., Cui, C., Ai, F. & Yin, D. Multi-generational obesogenic effects of sulfomethoxazole on *Caenorhabditis elegans* through epigenetic regulation. *Journal of hazardous materials* **382**, 121061, doi:10.1016/j.jhazmat.2019.121061 (2020).

R16. Chow, Y. L. & Sato, F. Transgenerational lipid-reducing activity of benzyloquinoline alkaloids in *Caenorhabditis elegans*. *Genes to Cells* **24**, 70-81 (2019).

R37. Zhao, Y. & Garcia, B. A. Comprehensive Catalog of Currently Documented Histone Modifications. *Cold Spring Harb Perspect Biol* **7**, a025064, doi:10.1101/cshperspect.a025064 (2015).

R38. Fullston, T. et al. Paternal obesity initiates metabolic disturbances in two generations of mice with incomplete penetrance to the F2 generation and alters the transcriptional profile of testis and sperm microRNA content. *FASEB J* **27**, 4226-4243, doi:10.1096/fj.12-224048 (2013).

R39. de Castro Barbosa, T. et al. High-fat diet reprograms the epigenome of rat spermatozoa and transgenerationally affects metabolism of the offspring. *Mol Metab* **5**, 184-197, doi:10.1016/j.molmet.2015.12.002 (2016).

3. Figure 1f and 1g: if the authors are examining for dosage effects that occur between populations

with varying numbers of generations receiving retreatments, an ANOVA should be performed to compare all the treatment groups. Just comparing a single treatment to controls will not detect these effects.

Our response: We thank the reviewer for his/her suggestion for statistical analyses, and have re-compared all the treatment groups using ANOVA test. These results also showed that the exposure of four consecutive generations of animals (P0, F1, F2, and F3), three (F1, F2, and F3), two (F2, and F3), or one (F3), did not affect lipid accumulation of their recovered F4 or F5 descendants. The corresponding results are shown in Fig. S1 f&g (the following graph) in revised manuscript.

4. I am not convinced about the role of SBP-1 in the TEI of lipid metabolism. For one, *sbp-1* mutants appear to have almost no stored lipids according to the ORO staining results compared to wild type, similar to what has been observed in other studies. Feeding *sbp-1* a high fat diet appears to rescue their lipid levels to wild type levels. A rescue of *sbp-1* stored lipid levels has previously been shown for oleic acid supplementation (Yang et al Nature 2006). How do you eliminate the possibility that these mutants are unable to store lipids from the possibility that they are regulating TEI phenotype? This question is also relevant for Figure 3 examining the fat mutants. The possibility exists that *sbp-1* and the fat mutants have “received” the signal for altered lipid storage, but are not capable due to the nature of their mutation. I do not think it is surprising fat storage mutants do not exhibit increased lipid storage phenotypes.

Our response: We thank the reviewer for raising this point. As the reviewer mentioned, consistent with previous reports, we also found that *sbp-1* mutants have almost no stored lipids, and feeding *sbp-1* HFD could rescue the lipid levels. It is not surprising to observe these phenotypes since SBP-1 is well-known for its function in regulating

lipid homeostasis. The reviewer is concerned that we might not be able to differentiate the role of SBP-1 in storing lipids from regulating TEI. Due to the nature of SBP-1, we could assume that SBP-1 can "execute" lipid accumulation by regulating the target genes, no matter whichever step they might be involved.

To our surprise, SBP-1 is involved in the "transmitting" step. This has been suggested by our data in Fig. 2h, in which RNAi-silencing of P0s' *sbp-1* could lead to the loss of F1s' (fed on OP50 and *sbp-1* was intact) ability to elevate lipid level. Here, the signal was initiated in HFD-fed P0 animals that had silenced *sbp-1*, but P0 couldn't pass down the signal to F1s, suggesting a role of *sbp-1* in transmitting TEI signal. Moreover, we also had data showing that *sbp-1* loss of function could block HFD-induced elevation of H3K4me3 (Fig.4e), indicating a regulating role of *sbp-1* in "transmitting" step.

For *fat* mutants, our data showed that they solely functioned as executors of fat accumulation. The biochemical function of FAT-5/6/7 can also explain their role in worms to accumulate lipids. We confirmed that FAT-5/6/7 could "execute" the accumulation of lipids, but not "transmit" the TEI signal in these processes.

5. *P10: The authors examine the potential role of chromatin modifiers in the regulation of their TEI phenotype. They cite a review and Han et al. to justify the link between the H3K4me3 chromatin modification and lipid metabolism. However, they examine the mutants met-1 (a H3K36me3 methyltransferase) and jmjd-3.1 (a H3K27me2/3 demethylase) which is not justified. In Han et al. they show that members of the COMPASS complex, which includes the SET-2 H3K4me3 methyltransferase, regulates lipid metabolism genes, and that other complexes do not appear to regulate these pathways. The authors show convincing data that WDR-5.1 plays a role in TEI of increased lipid storage. This section would be strengthened by also examining set-2 and ash-2 in their assay.*

Our response: We thank the reviewer for raising this point. We examined the role of *set-2* and *ash-2* in TEI of increased lipid storage, and found that neither *set-2* nor *ash-2* play a role in obesity TEI effect in our setting. The corresponding results are shown in Fig. S4a and b (the following graph) in revised manuscript.

6. *The correlation between H3K4me3 levels and TEI of lipid storage is not clear. First, H3K4me3 appears elevated in sbp-1 mutants in control and untreated samples compared to wild type, but do not show increased lipid storage (see comment above). This suggests that elevated H3K4me3 alone is not sufficient for this phenotype. Related to this, fat-5 and fat-7 genes are upregulated in their treatment animals (P0, F1, and F2), but H3K4me3 levels in their promoters are unchanged. Additionally, NHR-49 and NHR-80 are required for TEI, but these genes have no changes in gene expression or H3K4me3 levels. The authors should address these nuances in their data and explain a potential model for how H3K4me3 and lipid storage in more detail.*

Our response: We would like to thank the reviewer for pointing this out. We apologize for having presented confusing data (Fig. 4b and 4e in our previous version), which made the reviewer comment “*H3K4me3 appears elevated in sbp-1 mutants in control and untreated samples compared to wild type*”. As a matter of fact, H3K4me3 level is equivalent between *sbp-1* mutant and wild type in control and untreated samples, as shown in our revised manuscript (Fig.4e and S6a, see below). The previous images showing H3K4me3 (*sbp-1* and wild-type) blots were from two different films that had different exposure time, leading to an impression of different levels of protein expression. When we repeated these western blot experiments and run them in the same gel, the same exposure time gave us a similar protein level. Quantification analyses also confirmed that the levels of H3K4me3 of *sbp-1* and wild-type had no difference.

Furthermore, in our present research, we found that the mRNA levels of *fat-5* and *fat-7* genes were upregulated in their treatment animals (P0, F1, and F2), but H3K4me3 levels in their promoters are unchanged. Additionally, NHR-49 and NHR-80 are required for TEI, but these genes have no changes in gene expression or H3K4me3

levels. We revised our manuscript and clarified our explanation to present a potential model in our obesity TEI induced by HFD as:

“Upon animals being fed with HFD, the stress of lipid accumulation in the parental generation induces activity of lipid metabolic transcription factors *sbp-1*, *daf-16*, *nhr-49*, and *nhr-80*. At the same time, *sbp-1* regulates histone H3K4me3 modification, and establishes the epigenetic marks in descendants. In turn, the H3K4me3 marks in the progeny promote the recruitment of lipid metabolism-related genes (i.e. *sbp-1* and *daf-16*) and facilitate the activation; simultaneously, the activation of *daf-16* and *sbp-1* might recruit *nhr-49* or *nhr-80* to form complex, and then *daf-16*, *sbp-1*, *nhr-49* and *nhr-80* synergistically induce the expression of lipid metabolism-related genes (i.e. *fat-5*, *fat-6* and *fat-7*) to respond lipid metabolism and ultimately reset the metabolic processes, thereby completing the TEI of obesity effect (Fig 6).”

7. P6: very little attention is given to the RNA-seq data. As is, it adds little value to the results. What are the 25 genes that overlap with P0, F1, and F2 populations? In referring to these results, the manuscript states they are “consistent with a previous study”, but there is no citation. In addition, the methods should describe in detail how the RNA-seq libraries were analyzed.

Our response: We agree and thank the reviewer for this suggestion. We don’t see a need for these data and thus have entirely removed it. New data of CHIP-seq and RNA-seq were provided in our revision, and detailed methods were described following the reviewer’s advice.

8. Lipid metabolism and chromatin modifications are mechanistically linked to lifespan. In general, adults with increased lipid storage and decreased H3K4me3 levels have increased lifespans. Since treatment animals in this study exhibit increased lipid storage AND increased H3K4me3 levels, the lifespan prediction is unclear. The authors have an opportunity to tease apart these two pathways in their regulation of an important life history trait.

Our response: We thank the reviewer for raising this point. We performed the survival analyses, and found no difference between worms fed with HFD or OP50, suggesting that egg yolk did not affect survival of *C. elegans*. The corresponding results are shown in Fig. S10a-b (the following graph) in revised manuscript.

Minor concerns:

P3: “In modern societies, people no longer live in fear of famine. Instead, we live in a condition of over-nutrition...”. This statement is grossly incorrect. Many people around the world do not have food security. This sentence should be deleted.

Our response: We have deleted this sentence as suggested.

P6: “One concern is that HFD-induced fat accumulation could trigger a maternally heritable response in a dose-dependent manner “ Do the authors mean that the fat accumulated in the P0 is transmitted into F1 embryos via vitellogenesis, and that the extra lipids are maintained until adulthood and transmitted into the F2 embryos? This hypothesis seems highly unlikely, but should have a citation if it has been reported previously. I would recommend rewriting this part.

Our response: As suggested by the reviewer, we have re-written this part, where we say:

“In our heritable obesity effect model, we found that F2 populations exhibited a measurable but minor lipid accumulation phenotype. In order to find if a longer exposure might enhance F2s’ lipid accumulation, we exposed multiple consecutive generations of animals to HFD. We found that, the exposure of four consecutive generations of animals (P0, F1, F2, and F3), three (F1, F2, and F3),

two (F2, and F3), or one (F3), did not affect lipid accumulation their recovered F4 or F5 descendants (Fig 1f and g).”

P11: “Germline to Soma Communication” is not demonstrated by Figure 5.

Our response: We thank the reviewer for the suggestions. We have changed this section as “Characterization of tissue-specific functions of *daf-16* and *sbp-1* in HFD-induced TEI”

P10: “HFD-induced lipid accumulation in parents could just pass down to F2 generation, and F3 is back to normal lipid level, suggesting that HFD might induce epigenetic changes rather than genetic mutations in descendants” This is an odd statement to make towards the end of the results given how this phenotype has been labeled TEI from the beginning of the results.

Our response: We agree. We have deleted this sentence as suggested.

Figure 2: There are differences in wild type ORO staining in the treatment groups that are not explained.

Our response: We are sorry for a mistake here due to use segmented Y-axis and would like to thank the reviewer for pointing it out to us. We have revised this figure.

Reviewer #2 (Remarks to the Author):

In this study, Wan et al. showed that high fat diet induced lipid accumulation can pass to the next generations even they were fed with regular diet. They found that the transgeneration fat accumulation is regulated by transcription factors (DAF-16/FOXO and SBP-1), nuclear receptors (NHR-49 and NHR-80), and delta-9 desaturases. Furthermore, they demonstrated that the H3K4 trimethylation is enriched in the lipid metabolic genes, which promotes the multigeneration obesogenic effects. This study is novel and would be interesting to others in the community and the wider field. The data is convincing. Having said that, some questions need further address. First, the author performed RNAi and identified a list of genes up regulated by HFD. However, they did not go further to test them to see if any of them are required for transgenerational fat accumulation. Second, they showed many genes are required for the transgenerational fat accumulation, but the relationship among those genes has not been determined. For example, which genes initiate the signaling, how this signaling is passed to the others. From their data, it seems that DAF-16 acts upstream. But it did not affect H3K4me3, which suggests it is not required for the epigenetic changes required for the transgeneration effect. Could the author explain how DAF-16 acts as a “transmitter”?

Our response: We truly appreciate the reviewer’s positive feedback. For the questions mentioned by reviewer, we believe that understanding the relationship among those genes mediating multigeneration obesogenic effects induced by HFD will be an interesting topic. In our current manuscript, after charactering the roles of *wdr-5.1*, *daf-16*, *sbp-1*, *nhr-49*, *nhr-80*, *fat-5*, *fat-6* and *fat-7* in obesogenic TEI effects, we conducted two sets of RNAi experiments to definitively dissect whether the role of *daf-16*, *sbp-1*, *nhr-49*, *nhr-80*, *fat-5*, *fat-6* and *fat-7* is to implement regulation of lipid level (an “executor”), to transmit the heritable memories (a “transmitter”), or both. Our results demonstrated that *daf-16*, *sbp-1*, *nhr-49*, *nhr-80*, *fat-5*, *fat-6* and *fat-7* were required for HFD-induced TEI of lipid accumulation. Among them, *nhr-49*, *nhr-80*, *fat-5*, *fat-6*, and *fat-7* functioned solely as executors; *wdr-5.1* functioned solely as transmitter; *sbp-1* was responsible for transmitting the heritable memories, though we could not rule out its role as an executor; for *daf-16*, it not only functions as an executor to regulate lipid accumulation, but also as a transmitter to pass down heritable memory to progeny. Therefore, explain a potential model in our obesity TEI induced by HFD as: “Upon animals being fed with HFD, the stress of lipid accumulation in the parental generation

induces activity of lipid metabolic transcription factors *sbp-1*, *daf-16*, *nhr-49*, and *nhr-80*. At the same time, *sbp-1* regulates histone H3K4me3 modification, and establishes the epigenetic marks in descendants. In turn, the H3K4me3 marks in the progeny promote the recruitment of lipid metabolism-related genes (i.e. *sbp-1* and *daf-16*) and facilitate the activation; simultaneously, the activation of *daf-16* and *sbp-1* might recruit *nhr-49* or *nhr-80* to form complex, and then *daf-16*, *sbp-1*, *nhr-49* and *nhr-80* synergistically induce the expression of lipid metabolism-related genes (i.e. *fat-5*, *fat-6* and *fat-7*) to respond lipid metabolism and ultimately reset the metabolic processes, thereby completing the TEI of obesity effect (Fig 6).” However, for in-depth mechanism how DAF-16 act as a “transmitter”, we believe that it is an interesting topic for future characterization, and we will further reveal corresponding mechanism in further research.

Specific questions

1. *RNA seq data should be validated, either by quantitative rt-PCR or examining if they are involved in the fat accumulation in P0 or F1/F2.*

Our response: We agree and thank the reviewer for this suggestion. This is indeed a useful point and we actually confirmed RNA-seq data using rt-PCR. However, after careful consideration and combining the overall comments, we performed new experiments and provided new data of CHIP-seq and RNA-seq in our revision, we thus have entirely removed the previous RNA-seq section.

2. *In the figure 2, the authors showed nhr-49, nhr-80, sbp-1 and daf-16 are required for the transgeneration fat accumulation. However, the relationship among them is not clear. They should at least perform some epistasis analysis. For example, they could overexpress sbp-1/nhr-49/nhr-80 in the daf-16 mutant to see if it could suppress the daf-16 phenotype, or overexpressing daf-16 in other mutant background. Alternatively, they can also examine if the expression of those factors is regulated by others.*

Our response: The reviewer’s comments are really constructive and we have conducted corresponding analyses to clarify this problem as suggested.

To characterize the relationship of DAF-16, NHR-49, NHR-80 and SBP-1 in F1 generation, we perform epistasis analyses through overexpressing DAF-16 in the

background loss of *nhr-49*, *nhr-80* and *sbp-1*. We found that overexpressing DAF-16 in *nhr-49*, *nhr-80* and *sbp-1* mutant could not suppress the phenotype of *nhr-49*, *nhr-80* and *sbp-1* mutant. Here, we applied *sbp-1* RNAi in the background of overexpressing DAF-16 to analyze lipid level as we failed to obtain *daf-16* (OE); *sbp-1* double mutant by numerous mating. These results demonstrated that *daf-16*, *sbp-1*, *nhr-49* and *nhr-80* function in parallel pathway during the stress of lipid accumulation in F1 generation. The corresponding results are shown in Fig. S9 (the following graph) in revised manuscript.

3. The conclusion that *nhr-49* and *nhr-80* are purely executors based on the RNAi in P0. But it is also possible that the RNAi efficiency is low. They should exclude this possibility.

Our response: We thank the reviewer for the suggestions. Indeed, we paid attention to the RNAi efficiency during experiment, but we did not show the corresponding results in the initial manuscript for simplicity's sake. The corresponding results showed high knockdown efficiency as shown in the following graph.

4. *It is really interesting that they found DAF-16 is only required in muscle. But how does daf-16 sense the HFD? Is insulin or daf-2 involved?*

Our response: We thank the reviewer for pointing this out. We have additionally test whether the components of the insulin signaling pathway upstream of DAF-16 play a role to transmit signal. To determine this, we performed the lipid metabolism-associated germline or intestine-specific RNAi of *daf-2* (an insulin receptor-like gene) to detect the ORO staining. Our results exhibited that both germline and intestine specific RNAi of *daf-2* did not abrogate lipid accumulation induced by HFD, suggesting that DAF-16 senses other signals instead of the signals transmitted by *daf-2* to regulate lipid metabolism. The corresponding results are shown in Fig. S7d-f (the following graph) in revised manuscript.

5. *I expect that the executors for lipid accumulation in the P0 and F1/F2 would be different since they are fed with different diet. The authors should discuss this further.*

Our response: We thank the reviewer for the suggestions. We have included additional

discussion (Page 19, line 1-9), where we say,

“Because egg yolk is used as HFD to induce the phenotype of lipid aggregation, it is worth noting that the executors for lipid accumulation in the P0 and F1/F2 are different since they are fed with different diet. Therefore, in our present study, for P0 generation, only *daf-16*, but not *nhr-49*, *nhr-80*, *sbp-1*, *fat-5*, *fat-6* and *fat-7* was necessary as an executor to respond to lipid metabolism induced by egg yolk; while, for F1 generation, all of them (including *daf-16*, *nhr-49*, *nhr-80*, *sbp-1*, *fat-5*, *fat-6* and *fat-7*) are required as executors to respond to lipid accumulation.”

6. For data analysis, the number of replicates, the total number of animals should be indicated in the methods or figures. The authors normalized their data with *cdc-42*. Just curious, did they verified if *cdc-42* expression is affected by HFD? The immunoblot data should be quantified and analyzed. The data in the table S1 is not exactly correlated to the figure. For example, the figure 1a has five type of HFD treatment. Only one is showed in the table. And the description is not consistent. For example, in the figure, it shows P0-F5. In the table, it describes as P1, P4.

Our response: We thank the reviewer for these professional suggestions.

- 1) As suggested, the number of replicates and the total number of animals have been added.
- 2) For *cdc-42* as an internal reference, we have additionally verified whether expression of *cdc-42* is affected by HFD using other internal reference genes which were confirmed by David Hoogewijs et. al (Hoogewijs, David, et al. BMC molecular biology 9.1 (2008): 1-8). The results showed that the mRNA level of *cdc-42* is not affected by HFD. The corresponding results are shown in the following graph.

- 3) The quantification of immunoblot data have added as suggested.
- 4) We apologize for this inconsistency, and we have revised and carefully gone through the entire manuscript.

7. *Other interesting questions including it would be interesting to know, if the fat accumulation signaling is maternal, paternal or both. They have showed feeding HFD from birth to L4 is sufficient to induce the transgeneration effect, but what is the minimum duration and the time window for feeding with HFD to induce the transgeneration effect?*

Our response: We thank the reviewer's comments and conducted corresponding analyses to clarify these problems.

- 1) To determine the contributions of the sperm and oocytes in transmission of information of lipid accumulation, we conducted reciprocal matings and measured lipid levels of cross progeny. Our results showed that both female fed with HFD and male fed with HFD transmitted information of lipid accumulation to the next generation, suggested a role both sperm and oocytes in transgenerational obesity effect. The corresponding results are shown in Fig. 1h (the following graph a) in revised manuscript.
- 2) To determine how long a parent must be exposed to HFD to transmit lipid accumulation signal to progeny, we fed P0 generation animals with HFD at the L1 stage for 12 or 24 hours, and then transferred them to NGM plates with normal food. The results showed that the ORO levels were significantly increased in HFD-fed parents and in their recovered F1 progeny (although to a lesser extent than worms from parents fed with HFD for longer periods), but not in F2 progeny, suggesting that parents need to be fed HFD for enough long time before they can show a TEI phenotype. The corresponding results are shown in Fig. S1d-e (the following graph b and c) in revised manuscript.

Reviewer #3 (Remarks to the Author):

This manuscript started from an interesting phenotype that the high-fat diet (HFD) induces transgenerational lipid accumulation in C. elegans. The authors characterized a series of transcription factors involved in lipid metabolic regulation, including NHR-49, NHR-80, SBP-1, and DAF-16, and found that SBP-1 is more specifically responsible for transmitting effect between generations (transmitter), NHR-80 and NHR-49 mediate the lipid accumulation in the descendants (executor), while DAF-16 is not only required for the HFD-induced lipid accumulation, but also act as both an executor and transmitter. They further examined delta-9 desaturases, including fat-5, fat-6 and fat-7, and showed that fat-5 is an executor and mediates the lipid accumulation in the descendants. Furthermore, the authors found that HFD induces H3K4me3 in a sbp-1 dependent manner, and wdr-5.1 is specifically required for HFD-induced H3K4me3 and transmitting the effect between generations. Overall, the manuscript is well-written and easy to follow. The discovery will be interesting to a broad audience. To fully support the model, there are several major points that are required to be addressed.

Our response: We truly appreciate the reviewer's positive feedback.

Major points:

- 1. To distinguish an executor and a transmitter, the authors have designed experiments where RNAi feeding is applied in P0 but not F1 and the lipid phenotype is examined in F1. It is known that in some cases, RNAi knockdown can be passed from P0 to F1. This transgenerational RNAi effect can become a possible confounding factor for their experimental design. Given this effect varies among different RNAi clones, the authors should test whether gene knockdown occurs in F1 for all the RNAi clones.*

Our response: We thank the reviewer for the suggestions. Indeed, we paid attention to the RNAi efficiency during experiments, but we did not show the results in the manuscript for simplicity's sake. The corresponding results are shown in the following graph. We also tested whether gene knockdown occurred in F1 worms for all the RNAi clones, and found no RNAi transmission happened in our tested genes (see below).

2. The authors claimed that SBP-1 and DAF-16 can both function as transmitters. However, as shown in Figure 2h and k, the lipid accumulation in treated F1 is not suppressed with either *sbp-1* or *daf-16* P0 RNAi. Instead, the lipid level is increased in untreated F1. So the current results cannot support the claim.

Our response: We would like to thank the reviewer for pointing this out. We apologize for having labeled our data in a confusing manner (Fig. 2h, 2k in our previous version, see below), which made the reviewer comment “as shown in Figure 2h and k, the lipid accumulation in treated F1 is not suppressed with either *sbp-1* or *daf-16* P0 RNAi. Instead, the lipid level is increased in untreated F1”.

In fact, for untreated P0 and F1 worms, the reason why the lipid level was higher in F1 was that P0 animals fed with *sbp-1* or *daf-16* RNAi bacteria, while F1 animals fed with OP50. In other words, P0 was knockdown worms (low expression of *sbp-1* or *daf-16*) while F1 was wild-type (as measured by mRNA levels, see point 1 response).

We claimed that SBP-1 and DAF-16 can both function as transmitters, because we found the silencing of *daf-16* or *sbp-1* in P0 caused the loss of lipid accumulation of F1 (OP50-fed) progeny from parents fed with HFD.

In this experiment, we silenced *sbp-1* and *daf-16* exclusively in the P0 generation, and then analyzed the fat level of F1 generation raised on OP50. As reported in our recent study (Wan et al. 2021, *Science Advances*), if a gene is a transmitter, the silencing of that gene in the P0 generation will prevent the transmission of trans-generational memory, which will result in the elimination of the lipid accumulation of the F1

generation. Alternatively, if a gene only functions as an executor, the silencing of that gene in the P0 generation should only influence the lipid level of P0 generation, so that we would still detect the elevation of fat level in F1 generation.

3. *Given the transgenerational effect does not pass to F3, the level of H3K4me3 should be also examined in F2 and F3. Based on the current model, the induction of H3K4me3 should be reduced over generations.*

Our response: We thank the reviewer for raising this point. We have examined the level of H3K4me3 in F2 and F3 generation. The corresponding results are shown in Fig. S4e-f (the following graph) in revised manuscript.

4. *Inactivation of wdr-5.1 decreases the H3K4me3 level, which is the opposite to the HFD treatment. However, the wdr-5.1 mutant has increased lipid accumulation. Whether H3K4me3 also targets genes that trigger lipid catabolism? Are any of these genes found in the ChIP-seq data?*
5. *Comparison of RNA-seq and ChIP-seq data should be conducted to provide a systemic view on the the transcriptional landscape.*

Our response (point 4 and 5): We thank the reviewer for these professional suggestions. We have additionally conducted CHIP-seq analysis as suggested. From CHIP-seq, we found that H3k4me3 marks some lipid metabolism-associated gene, and H3K4me3 modification of some of them was significantly up-regulated when worms fed with HFD. GO analysis showed most overlapping genes that were significantly changed in both CHIP-seq and RNA-seq data involved in lipid metabolic process. The corresponding results are shown in Fig. S5c-i (the following graph) in revised manuscript.

6. To further strengthen the connection between *SBP-1*, *H3K4me3* and the lipid phenotype, tissue-specific knockdown of *sbp-1* should be examined for the lipid phenotype in addition to the *H3K4me3* level. Given that similar studies have been done for *daf-16* (fig. 5g-i), there should not be technical difficulties.

Our response: As suggested by the reviewer, we have additionally conducted the tissue-specific RNAi of *sbp-1* to detect the ORO staining. Consistent with the results from *sbp-1* lof mutant, knockdown of *sbp-1* in different tissue could not abrogate lipid accumulation induced by HFD. The corresponding results are shown in Fig. S8 (the following graph) in revised manuscript.

7. Although inactivation of all delta-9 desaturases suppresses lipid accumulation in F1 and F2, only *fat-5* is tested using the executor/transmitter distinguishing experiments. To support the claim that delta-9 desaturases are executors, the authors should also test *fat-6* and *fat-7*.

Our response: We thank the reviewer for raising these points. We have additionally dissected roles of *fat-6* and *fat-7* using the executor/transmitter distinguishing experiments as suggested. The corresponding results are shown in Fig. S2f-h (the following graph) in revised manuscript.

8. Given that *DAF-16* is both an executor and transmitter, does *daf-16* knockdown affect *H3K4me3* levels? Also, is *Daf-16* nuclear translocation induced by HFD and transmitted to descendants? If yes, whether *DAF-16* nuclear translocation specifically occurs in the muscle?

Our response: We thank the reviewer for these professional suggestions. In our initial manuscript, we have shown that the change of H3K4me3 induced by HFD was not affected by DAF-16, as shown in Supplementary Figure S6b (formerly Supp. Fig S5b). Additionally, we have detected DAF-16 nuclear translocation and determined protein levels in animals fed with HFD. Indeed, we found that DAF-16 displayed obvious aggregation in the nucleus, especially in muscle, in the P0 generation fed with the HFD and in their recovery F1 progeny (Fig S7a). Moreover, we found that protein levels of DAF-16 were obviously elevated in P0 worms fed with HFD and their recovered F1 progeny. The corresponding results are shown in Fig. S7b-c (the following graph) in our revised manuscript.

Minor points:

1. Figure 2h: the scale of the y axis is not correct.

Our response: We apologize for this mistake and have corrected this figure. Further, we have now carefully gone through the entire manuscript and all figures.

REVIEWERS' COMMENTS

Reviewer #1 (Remarks to the Author):

In this revised manuscript entitled "Histone H3K4me3 modification is a transgenerational epigenetic signal for lipid metabolism in *Caenorhabditis elegans*", Wan et al. describe investigate the mechanism of transgenerational inheritance of obesity predisposition from parents fed a high fat diet. High fat diets in mammals have been shown to affect metabolic states transgenerationally, but the mechanisms of how a nutritional memory is transmitted across generations are not well understood. In this manuscript, the authors feed *C. elegans* hermaphrodites egg yolk as a model of high fat diet, which results in an increase in intestinal lipids and global histone H3K4me3 marks in the treated animal that is inherited for two generations. They found that transgenerational inheritance of high fat diet induced phenotypes required DAF-16/FOXO, SBP-1/SREBP, the delta-9 desaturases, fat-5/-6/-7, nuclear hormone receptors NHR-49 and NHR-80, as well as the histone methyltransferase WDR-5.1. This work is novel and would be of interest to the field. In addition, the authors made significant efforts to address my concerns, including performing new experiments and editing the manuscript.

I have only minor comments regarding the current manuscript

1. The authors should move all results from the discussion section to the results section.
2. The authors conducted mating experiments to test if the effects of the high fat diet could be passed from the male as well as the hermaphrodite. The strain used for this experiment was not stated, so that I would assume it was using N2 Bristol. How did the authors identify the cross progeny? Not all mating assays are 100% successful and only a portion of the progeny will be a result of cross fertilization. These results might not be valid if cross progeny were not separated from selfed progeny and tested for TEI.

Reviewer #2 (Remarks to the Author):

The authors addressed all questions I raised properly and I don't have any more questions. Therefore, I endorse the publication.

Reviewer #3 (Remarks to the Author):

The authors have addressed almost all of my comments, except for #6. Given that the *sbp-1* knockdown doesn't the HFD-induced lipid accumulation phenotype in P0, I was not suggesting to test the tissue-specificity of *sbp-1* knockdown in P0. Instead, I was suggesting the reviewer to test which tissue is crucial for the *sbp-1* knockdown to affect the transgenerational effect. I think this data will provide additional insight in understanding the transgenerational regulation by SBP-1, but this is not necessary for the manuscript to be published at Nature Communication. Therefore, I will recommend to accept the manuscript for publication.

Reviewer #1 (Remarks to the Author):

In this revised manuscript entitled “Histone H3K4me3 modification is a transgenerational epigenetic signal for lipid metabolism in Caenorhabditis elegans”, Wan et al. describe investigate the mechanism of transgenerational inheritance of obesity predisposition from parents fed a high fat diet. High fat diets in mammals have been shown to affect metabolic states transgenerationally, but the mechanisms of how a nutritional memory is transmitted across generations are not well understood. In this manuscript, the authors feed C. elegans hermaphrodites egg yolk as a model of high fat diet, which results in an increase in intestinal lipids and global histone H3K4me3 marks in the treated animal that is inherited for two generations. They found that transgenerational inheritance of high fat diet induced phenotypes required DAF-16/FOXO, SBP-1/SREBP, the delta-9 desaturases, fat-5/-6/-7, nuclear hormone receptors NHR-49 and NHR-80, as well as the histone methyltransferase WDR-5.1. This work is novel and would be of interest to the field. In addition, the authors made significant efforts to address my concerns, including performing new experiments and editing the manuscript.

Our response: We truly appreciate the reviewer’s positive feedback.

I have only minor comments regarding the current manuscript

1. The authors should move all results from the discussion section to the results section.

Our response: All results in discussion section have moved to the results section, as suggested.

2. The authors conducted mating experiments to test if the effects of the high fat diet could be passed from the male as well as the hermaphrodite. The strain used for this experiment was not stated, so that I would assume it was using N2 Bristol. How did the authors identify the cross progeny? Not all mating assays are 100% successful and only a portion of the progeny will be a result of cross fertilization. These results might not

be valid if cross progeny were not separated from selfed progeny and tested for TEI.

Our response: I apologize for missing strain names used in these mating experiments. In fact, to distinguish the cross progeny from the self-fertilized ones, we utilized GFP⁺ strain MT18143 males to cross with N2 Bristol female, and then picked GFP offspring to perform ORO staining experiment. Detailed mating protocol and strain MT18143 information have added in our revised manuscript.

Reviewer #2 (Remarks to the Author):

The authors addressed all questions I raised properly and I don't have any more questions. Therefore, I endorse the publication.

Our response: we are pleased that our revisions resolved the reviewer's concern.

Reviewer #3 (Remarks to the Author):

*The authors have addressed almost all of my comments, except for #6. Given that the *sbp-1* knockdown doesn't the HFD-induced lipid accumulation phenotype in P0, I was not suggesting to test the tissue-specificity of *sbp-1* knockdown in P0. Instead, I was suggesting the reviewer to test which tissue is crucial for the *sbp-1* knockdown to affect the transgenerational effect. I think this data will provide additional insight in understanding the transgenerational regulation by SBP-1, but this is not necessary for the manuscript to be published at Nature Communication. Therefore, I will recommend to accept the manuscript for publication.*

Our response: We thank the reviewer for this positive feedback. For suggestion “to understand which tissue is crucial for the *sbp-1* knockdown to affect the transgenerational effect” from the reviewer, we believe that it is an interesting and crucial topic and we will dissect corresponding mechanism in our future study.